# A rapid rise in hormone receptor-positive and HER2-positive breast cancer subtypes in Southern Thai women: A population-based study in Songkhla

**Aungkana Chuaychai**[1,2], **Hutcha Sriplung**[1]*

1 Department of Epidemiology, Faculty of Medicine, Prince of Songkla University, Songkhla, Thailand,
2 School of Pharmacy, Walailak University, Nakhon Si Thammarat, Thailand

* hutcha.s@psu.ac.th

**Data Availability Statement:** All S1 Dataset.Rdata, S2 Appendix.docx, and S2 Dataset.csv files are available from the DANS database at: https://doi.org/10.17026/dans-xn5-5286.

## Abstract

The incidence of breast cancer is increasing in low- and middle-income countries, including Thailand. However, its molecular immunohistochemical (M-IHC) subtypes have not been summarized in a population-based cancer registry. Thus, we aimed to estimate the breast cancer incidence and trends based on the hormone receptor and human epidermal growth factor receptor 2 (HER2) status. This cross-sectional study included 2,883 women diagnosed with primary invasive breast cancer between 2009 and 2018 from the Songkhla Cancer Registry. After imputing the missing values of estrogen receptor (ER), progesterone receptor (PR), and HER2 status, the cases were classified into four subtypes: HR+/HER2−, HR+/HER2+, HR−/HER2−, and HR−/HER2+. The age-specific incidence rate of 5-year age groups and age-standardized incidence rate (ASR) were calculated. An age-period-cohort (APC) model was used to describe the effects of age, birth cohort, and period of diagnosis. Finally, the incidence trends were extrapolated to 2030 based on the APC and joinpoint models. The results showed, HR+/HER2− had the highest ASR in breast cancer. The incidence trends of HR+/HER2− and HR+/HER2+ increased with an annual percent change of 5.4% (95%CI: 2.5% to 8.3%) and 10.1% (95%CI: 4.9% to 15.5%), respectively. The rate ratio was high in the younger generation and recent period of diagnosis. The joinpoint and APC model projections showed that the ASR of HR+/HER2− would reach 30.0 and 29.2 cases per 100,000 women, while ASR of the HR+/HER2+ would reach 8.8 and 10.4 cases per 100,000 women in 2030. On the other hand, the incidence trends of the HR−/HER2− and HR−/HER2+ subtypes were stable. The rising trends of HR-positive and a part of HER2-positive breast cancer forecast a dynamicity of the future health care budgeting, resource allocation, and provision of facilities.

**Funding:** The author(s) received no specific funding for this work.

**Competing interests:** The authors have declared that no competing interests exist.

## Introduction

Breast cancer is the most common cancer among women worldwide [1]. In 2020, the age-standardized incidence rate (ASR World) estimated by GLOBOCAN was 47.8 cases per 100,000 women, while the number of female patients with breast cancer was nearly a quarter of the total proportion of women with various types of cancers worldwide [1, 2]. The incidence of breast cancer was relatively high and stable in countries with very high and high human development index (HDI) scores, respectively [1]. By contrast, countries with medium and low HDI scores had a lower incidence [1], while the cancer trends increased [1].

The incidence of breast cancer has been rising in Thailand, especially among women aged >50 years [3, 4]. The proportion of luminal A-like subtype was significantly high among the molecular immunohistochemical (M-IHC) subtypes of breast cancer [5]. Nevertheless, triple-negative cancer was disproportionally high among Muslim women in Southern Thailand. Moreover, Muslim women with breast cancer, showed poor survival [6].

Since 2002, Thailand has implemented a universal health coverage (UHC) scheme [7]. This is the primary insurance scheme for Thai citizens. As of 2020, the UHC scheme has already provided health services to approximately 79% of the Thai population [8]. It covers health promotion, disease screening, treatment of basic and high-cost diseases under the Thai National List of Essential Medicines (NLEM), and palliative/supportive care.

The main goal of breast cancer treatment is to cure the disease. Surgery is the primary option for the tumor removal, while adjuvant therapy destroys the residual malignant cells. The following four main types of treatment modalities are used for breast cancer treatment: chemotherapy, hormonal therapy, targeted therapy, and radiotherapy. The use of hormone and targeted therapy is dependent on the female sex hormone receptor (HR) (estrogen receptors [ER] and progesterone receptors [PR]) and human epidermal growth factor receptor 2 (HER2) status in addition to the histologic type of breast cancer, grade, lymph nodes, and distant metastatic organ involvement [9, 10].

Trastuzumab (anti-HER2) was included in the NLEM in 2014 [11]. At a certified cancer clinic, patients with breast cancer who require targeted therapy can be registered in the national medical insurance funds and get access to the drug. Trastuzumab has proven to be cost-effective in the treatment of patients with early-stage breast cancer in Thailand [11]. Although targeted therapy is a cost-effective treatment for early-stage breast cancer, it costs 15,560 USD per patient to add a targeted therapy to the baseline treatment; meanwhile, the quality-adjusted life years (QALYs) expected to increase by 4.59 years [11]. Thus, the national health budget increases as the number of patients with early-stage breast cancer increase. However, longer life gain for women who benefited from a targeted therapy would make a return in terms of the gross domestic product (GDP) per capita, the benefit of which is greater than the budget spent by the UC scheme for the cost of treatment per patient.

Other targeted drugs approved by the Thai Food and Drug Administration for breast cancer treatment include pertuzumab and ribociclib. Currently, these are the only two drugs available to patients who are covered by the civil servant medical benefit scheme [12, 13]. Drug indications mainly depend on the HR and HER2 status. Therefore, determining the actual proportion and trends of breast cancer according to the HR and HER2 status is necessary. This data can help predict the need for targeted drug therapy as well as the survival in each breast cancer subtype. It also provides local information that helps increase the healthcare planning validity and assists in the proper allocation of healthcare resources.

The M-IHC subtype is not a mandatory reportable type of breast cancer according to the World Health Organization criteria. However, it is preferable for all cancer registries to record and report the incidence of M-IHC subtypes. The existing system of Songklanagarind Hospital

reports both population-based and hospital-based statistics for histological but not M-IHC subtypes. The HR and HER2 status can be retrieved from the hospital information system and the pathology records to estimate the ASR and incidence trends based on the M-IHC subtypes of breast cancer. Thus, to guide breast oncologists and health insurance scheme decision-makers, this study aimed to estimate the breast cancer incidence and trends based on the HR and HER2 status. Furthermore, we aimed to project the number of cancer cases and the required expansion of healthcare budget for trastuzumab treatment in 2030.

## Materials and methods

### Study design and participants

This was a population-based study. The protocol was approved by the Human Research Ethics Committee, Faculty of Medicine, Prince of Songkhla University (REC.63-031-18-1). The requirement for obtaining an informed consent was waived by the ethics committee as the research poses no more than minimal risk to the participants, and the researchers agreed not to disclose the personal identification to third parties. Only patient identification information, which included hospital numbers, was used in this study to merge participants' data files from two data sources. In the analysis and study report, all data was fully anonymous.

All patients with primary invasive breast cancers who were diagnosed between January 2009 to December 2018 were included in this study. The breast cancer cases were identified from the Songkhla Cancer Registry, a provincial population-based cancer registry, based on the 10th version of the International Classification of Diseases codes combined with the ICD, Oncology version 3 codes. The topography code for breast cancer (C50.x) and behavior code for malignancy (code 3) were selected. Patients with phyllodes tumors (morphology code: 9020) were excluded.

### Study setting and data sources

Songkhla province is located in Southern Thailand, between 6°17′-7°56′ N latitude and 100°1′-101°6′ E longitude. The province has an area of approximately 7,394 square kilometers. As of 2019, Songkhla has a total population of 1.4 million people, of which 0.7 million (51.2%) are women [14].

The information on patients with breast cancer was collected from two data sources: the Songkhla Cancer Registry database and medical records of patients in Songklanagarind Hospital. The cancer registry staff extracted the data from the cancer registry database on March 25, 2020, and the researchers collected the data from the patient's medical records from April 20, 2020, to October 18, 2020.

The Songkhla Cancer Registry is a population-based cancer registry that enrolled cancer patients residing in Songkhla Province. Case ascertainment began in 1988, and patients' data were regularly supplied to the Cancer Incidence in Five Continents (CIV since vol. VIII) [15]. The cases were identified using active and passive case-finding methods and registered by trained staff. The sources of information included medical records and pathological records of all hospitals in the province. The vital status of the patients in the database is reported when deaths occur in the hospitals and healthcare network and is confirmed by the Bureau of Registration Administration Database, Department of Provincial Administration, Ministry of Interior.

The date of incidence was obtained from the most reliable source. The incidence date was firstly indicated as the date of biopsy, if not available, the date when the specimen was received at the pathological laboratory, or the report date. When the date of definitive pathological diagnosis was not indicated, the date of hospital admission due to this malignancy, radiological

diagnosis, and other clinical diagnosis dates related to the occurrence of malignancy or death, death certificate notification only (DCO) were considered. The rules for reporting the incidence date were according to the European Network of Cancer Registries (ENCR) [16], which was also used by the Thai Cancer Registry Network and the International Agency for Research on Cancer (IARC). The IARC allowed missing pathological diagnosis in capturing cancer cases for population-based cancer registration as a low percentage of patients migrating out of the captive area of a cancer registry usually occurred in long-surviving diseases and death might occur before diagnosis in very short surviving diseases.

Songklanagarind Hospital is a super-tertiary hospital in Songkhla Province under the Prince of Songkla University. It has sophisticated diagnostic and therapeutic medical equipment and is the referral center for cancer patients, including those with breast cancer from other hospitals in Songkhla and nearby provinces. In the present study, we included only cases living in Songkhla province and registered in the population-based cancer registry.

## Study variables

Most of the variables in this study were extracted from the Songkhla Cancer Registry database. The variables included patient's demographic data and tumor characteristics. Data on the ER, PR, and HER2 status were collected from the electronic medical records of the Songklanagarind Hospital.

## Immunohistochemical staining

The histopathology laboratory of the Songklanagarind Hospital interpreted the HR and HER2 status following the American Society of Clinical Oncology/College of American Pathologists guidelines. Invasive tumors with at least 1% positive nuclei staining for ER or PR were considered positive for HR.

The HER2 status is reported into three groups based on the immunohistochemical (IHC) staining levels: negative, equivocal, and positive. In this study, the equivocal and negative groups were combined into one category.

## Population denominators

The Songkhla population projection was used as the population denominator to compute the age-specific incidence rate and ASR from 2010 to 2018. The expected populations were reported by the Office of the National Economic and Social Development Board of Thailand [17]. However, the report did not include population projections of 2009. Hence, we estimated the population denominator of 2009 based on the population censuses in 2000 and 2010, using a log-linear function. The two consecutive censuses were published by the National Statistical Office [18]. The projection data was deposit in the Data Archiving and Networking Services (DANS): https://easy.dans.knaw.nl/ui/datasets/id/easy-dataset:230753 [19].

## Statistical analysis

**Incidence rates, trends, and projection.** To estimate the incidence rates of breast cancer, the disease was classified into four subtypes by performing an M-IHC: 1) ER- or PR-positive and HER2 negative (HR+/HER2−, luminal A-like), 2) ER- or PR-positive and HER2 positive (HR+/HER2+, luminal B-like), 3) ER- and PR-negative, and HER2 negative (HR−/HER2−, triple-negative) and 4) ER- and PR-negative and HER2 positive (HR−/HER2+, HER2-enriched). The age-specific incidence rates in each 5-year age group (0–4, 5–9, 10–14,. . ., 80–84, and ≥85

years) were calculated using the estimated Songkhla population (as described above), while the ASR was calculated using the world standard population [20] proposed by Segi (1960) and modified by Doll et al. (1966) following the direct method. The count of cases/ estimated Songkhla population in each 5-year age group might result in the age-specific rate of zero in very young age groups.

The effects of age, birth cohort, and year of diagnosis (period) were assessed by fitting a log-linear model to the observed data, assuming a Poisson distribution. The two Poisson models, AP-C and AC-P, where the cohort or period effects were modeled with age-period or age-cohort pairs, were set as an offset, and used to describe the impact of the time component on the incidence trends.

The joinpoint regression analysis was performed to identify the change in incidence trend [21]. The changes in incidence trends over time were described as annual percent changes.

The projection of the incidence rate was extrapolated to 2030 using the joinpoint and age-period-cohort (APC) models. In the joinpoint model, future were projected trends based on the estimated regression coefficients from 2009 to 2018. At the same time, linear interpolation was used to estimate the cohort and period effects of the APC model, which were extrapolated to the future. First, the age-specific incidence rates in each projection year were calculated using age, projected cohort, and projected period effect. Then, if the projected rates deviated from the overall trends, the average was used to smooth the outliers. Subsequently, the ASR of the 5-year interval was computed using the modified Segi world population based on the age-specific incidence rates of the middle-age group in each interval. In addition, the Holford method was used to extract the linear drift from the APC model. Finally, the cut trend concept was applied to our projection by 0% attenuation trends in the first 5 projection years, following the geometric dampening with the factor of 0.92 (1–0.08) per year both in the joinpoint and the APC models [22, 23].

**Trastuzumab cost projection.** The incremental cost of trastuzumab treatment was projected in 2030 based on three sources of information, including our projection of age-specific incidence rates of the HER2-positive subtype in 18 age groups (0–4, 5–9, 10–14,. . ., 80–84, and ≥85 years), Thai population projection of 18 age groups in 2030 [17], and pharmacoeconomic evaluation of trastuzumab treatment in Thailand [11].

Based on the APC model mentioned in the previous section, the age-specific incidence rates were projected in 2030. The total estimated number of new HER2-positive cases in 2030 referred to the summation of the new cases in each age group, calculated by multiplying the rate by the projection of Thai women in each stratum.

Then, the maximum number of eligible cases for trastuzumab treatment was estimated based on the proportion of patients with early-stage breast cancer, including those with local and regional stage, in our imputed datasets. Finally, the required additional budget in 2030 due to trastuzumab treatment was approximate based on the results of the budget impact analysis from the pharmacoeconomic study of trastuzumab treatment in Thailand. The results showed that the incremental cost was 475,921 THB per case per year, which was approximately 13,998 USD (1 USD = 34 THB) in 2021.

## Management of missing data

Since the HR and HER2 status were not routinely reported together with the histopathological diagnosis of all breast cancer cases, we imputed the missing values of the receptor status using the multivariate imputation by chained equations (MICE) package in R-program [24]. As we predicted the receptor status variables to create the four subtypes of breast cancer, other demographic variables, including age, religion, year of diagnosis, and

histopathologic variables, including morphology, grade and stage of tumor, were used to impute the missing HR and HER2 variables. The missing values in the variables were explored other than receptor status, such as grade and stage of the disease, and were found to be less than 22%. Thus, the missing values were set as the "unknown" class in the variables so that the "unknown" values could be used to predict the receptor status in the imputation. This process could avoid unclearly defined strata of the predictor variables, while the missing values in the predictor variables were also imputed during the cycles of chained equations. In addition, the values of one of the three receptor status were used to predict the missing values of the other missing receptor status. The imputation link was set to the logistic regression (logreg) model, and 1,000 datasets were generated. Some of them were known values from the original data, and the other were imputed data filled into the missing (blank) values. The dataset after imputation was deposited in the Data Archiving and Networking Services (DANS): https://easy.dans.knaw.nl/ui/datasets/id/easy-dataset:230753 [19].

The MICE package does not calculate the confidence interval (CI) around the mean of the imputed results for the outcomes of the categorical variables. Therefore, the percentile of the imputed results of the HR status and HER2 receptor status were computed; the results showed that the 2.5th and 97.5th percentiles simulated the 95% probability interval.

## Results

### Participants and characteristics

Among 2,909 women with breast cancer registered in the Songkhla Cancer Registry, 12 cases with in-situ tumors and 14 cases with phyllodes tumors were excluded. Hence, only 2,883 patients (99.1% of the initial dataset) with breast cancer diagnosed from January 2009 to December 2018 were included in the study.

The median age of the patients at diagnosis of breast cancer was 53.0 years (interquartile range, IQR: 46.0–62.0; range: 24–95), and most of them were diagnosed with regional tumors (57.2%). Ductal carcinoma was the most common tumor type found in 82.4% of the cases; meanwhile, lobular carcinoma was the second most common tumor type and was found in 4.6% of the cases.

HR status was reported in 54.4% of the patients. Of them 1,098 (38.1%) and 883 (30.6%) had ER-positive status and PR-positive status, respectively. HER2 status was reported in approximately half of the patients (50.2%). Positive HER2 status was observed in 319 (11.1%) patients, while the equivocal/negative status was reported in 1,127 (39.1%) patients. In the imputed data were 69.3% and 56.2% of the patients had ER-positive status and PR-positive status, respectively, while only 23.2% of patients had HER2-positive status (Table 1, S1 and S2 Tables).

### Proportions of breast cancer M-IHC subtypes

The proportions and confidence/probability intervals of the observed and imputed data were comparable for each breast cancer subtype. The HR+/HER2− or luminal A-like subtype was the predominant subtype, accounting for approximately 60% of all cases. The second most common subtype, HR−/HER2− or triple-negative, comprised 17% of all the cases, as shown in Table 2. The subtypes in which the proportion was computed from the observed data tended to regress towards 0.5 (50%) after the imputation process. The estimates and margin of error of the imputed data were calculated using three methods: median and 95% probability interval, mean and Rubin's estimate of the 95% CI, and mean ±2 standard deviation margins.

**Table 1. Demographic and tumor characteristics of observed and imputed data.**

| Variables | Observed data (all = 2883) n (%) | Imputed data (all = 2883) n (%) |
|---|---|---|
| **Age** (median: 53; IQR: 46–62; range: 24–95) | | |
| 20–29 | 27 (1.0) | 27 (1.0) |
| 30–39 | 269 (9.3) | 269 (9.3) |
| 40–49 | 807 (28) | 807 (28) |
| 50–59 | 874 (30.3) | 874 (30.3) |
| 60–69 | 545 (18.9) | 545 (18.9) |
| 70 and over | 361 (12.5) | 361 (12.5) |
| **Religion** | | |
| Buddhist | 2508 (87) | 2508 (87) |
| Muslim | 350 (12.1) | 350 (12.1) |
| other | 8 (0.3) | 8 (0.3) |
| unknown | 17 (0.6) | 17 (0.6) |
| **Morphology** | | |
| ductal | 2376 (82.4) | 2376 (82.4) |
| lobular | 134 (4.6) | 134 (4.6) |
| mixed | 70 (2.4) | 70 (2.4) |
| others | 93 (3.2) | 93 (3.2) |
| unknown | 210 (7.3) | 210 (7.3) |
| **Grade** | | |
| well-differentiated | 307 (10.6) | 307 (10.6) |
| moderately- differentiated | 1072 (37.2) | 1072 (37.2) |
| poorly- differentiated | 861 (29.9) | 861 (29.9) |
| undifferentiated | 8 (0.3) | 8 (0.3) |
| unknown | 635 (22) | 635 (22) |
| **Stage** | | |
| local | 449 (15.6) | 449 (15.6) |
| regional | 1648 (57.2) | 1648 (57.2) |
| distant | 243 (8.4) | 243 (8.4) |
| unknown | 543 (18.8) | 543 (18.8) |
| **Estrogen receptor** | | |
| negative | 470 (16.3) | 886 (30.7) |
| positive | 1098 (38.1) | 1997 (69.3) |
| unknown | 1315 (45.6) | |
| **Progesterone receptor** | | |
| negative | 684 (23.7) | 1264 (43.8) |
| positive | 883 (30.6) | 1619 (56.2) |
| unknown | 1316 (45.6) | |
| **HER2-receptor** | | |
| negative/equivocal | 1127 (39.1) | 2214 (76.8) |
| positive | 319 (11.1) | 669 (23.2) |
| unknown | 1437 (49.8) | |
| **Year at diagnosis** | | |
| 2009 | 206 (7.1) | 206 (7.1) |
| 2010 | 227 (7.9) | 227 (7.9) |
| 2011 | 293 (10.2) | 293 (10.2) |
| 2012 | 221 (7.7) | 221 (7.7) |
| 2013 | 269 (9.3) | 269 (9.3) |

(*Continued*)

**Table 1.** (Continued)

| Variables | Observed data (all = 2883) n (%) | Imputed data (all = 2883) n (%) |
|---|---|---|
| 2014 | 291 (10.1) | 291 (10.1) |
| 2015 | 296 (10.3) | 296 (10.3) |
| 2016 | 354 (12.3) | 354 (12.3) |
| 2017 | 360 (12.5) | 360 (12.5) |
| 2018 | 366 (12.7) | 366 (12.7) |

## Age-specific incidence rates

Fig 1 shows the age-specific incidence rates per 100,000 women according to the M-IHC subtype. The incidence of all subtypes tended to increase in the 20–24-year age group. The highest peak of the age-specific incidence was observed in the HR+/HER2− (luminal A-like) subtype, reaching 60.2 cases per 100,000 women at the age of 61.4 years (95%CI: 52.3 to 70.6); followed by the HR−/HER2− (triple-negative) subtype with 17.2 cases at the age of 60.4 years (95%CI: 48.9 to 72.0), and the HR−/HER2+ (HER2-enriched) subtype with 13.4 cases at the age of 66.2 years (95%CI: 59.8 to 72.5); meanwhile, the lowest peak was observed in the HR+/HER2+ (luminal B-like) subtype, with 11.3 cases per 100,000 women at the age of 58.7 years (95%CI: 51.1 to 66.3). The rates declined in all subtypes in the older age groups after the age of 70 years. Thus, the incidence of the HR−/HER2+ subtype peaked later than the other three subtypes.

## Age-cohort-period analysis of the age-standardized incidence rates

Fig 2 shows the APC analysis of breast cancer incidence by M-IHC subtype. The HR+/HER2− (luminal A-like) and HR+/HER2+ (luminal B-like) subtypes were affected by the birth cohort and time of diagnosis. In the AC-P models (blue lines in Fig 2A and 2B), the younger birth

**Table 2.** The proportion of breast cancer subtypes and confidence/probability intervals among the observed and imputed datasets.

| Breast cancer Subtypes | Observed data | | Imputed data[C] | | | |
|---|---|---|---|---|---|---|
| | Number of cases[A] (%) n = 2883 | Proportion[B] (95%CI)[D] | Median number of cases (%); (95% PI)[E, F] | Median proportion and 95%PI[E] (Quantile) | Mean proportion and 95%CI[G] (Rubin's) | Mean proportion ±2 SD |
| HR+/HER2− (luminal A-like) | 880 (30.5) | 0.609 (0.584 to 0.635) | 1718(59.7); (1642–1789) | 0.596 (0.570 to 0.621) | 0.596 (0.571 to 0.620) | 0.596 (0.570 to 0.621) |
| HR+/HER2+ (luminal B-like) | 150 (5.2) | 0.104 (0.088 to 0.120) | 318(11.0); (277–374) | 0.110 (0.096 to 0.130) | 0.111 (0.094 to 0.127) | 0.111 (0.094 to 0.128) |
| HR−/HER2− (triple-negative) | 246 (8.5) | 0.170 (0.151 to 0.190) | 497 (17.3); (447–552) | 0.172 (0.155 to 0.191) | 0.172 (0.154 to 0.191) | 0.172 (0.154 to 0.191) |
| HR−/HER2+ (HER2-enriched) | 168 (5.8) | 0.116 (0.100 to 0.133) | 347(12.0); (306–400) | 0.120 (0.106 to 0.139) | 0.121 (0.104 to 0.138) | 0.121 (0.104 to 0.138) |

[A] 1,439 subjects with unknown subtype,

[B] proportion excluding the unknown subtype,

[C] subtypes estimation based on the known cases and the imputed receptor status cases,

[D] 95% CI is the confidence interval estimated from the proportion ± $Z_{\alpha/2}$ x standard error, where α is 0.05,

[E] 95% PI is the probability interval estimated from 0.025 to 0.975 quantiles.

[F] total cases are not equal to 2,883 cases because it's the average number from 1,000 datasets

[G] 95% CI is the confidence interval estimated by Rubin's rule for mean (Show the detail in S1 Appendix)

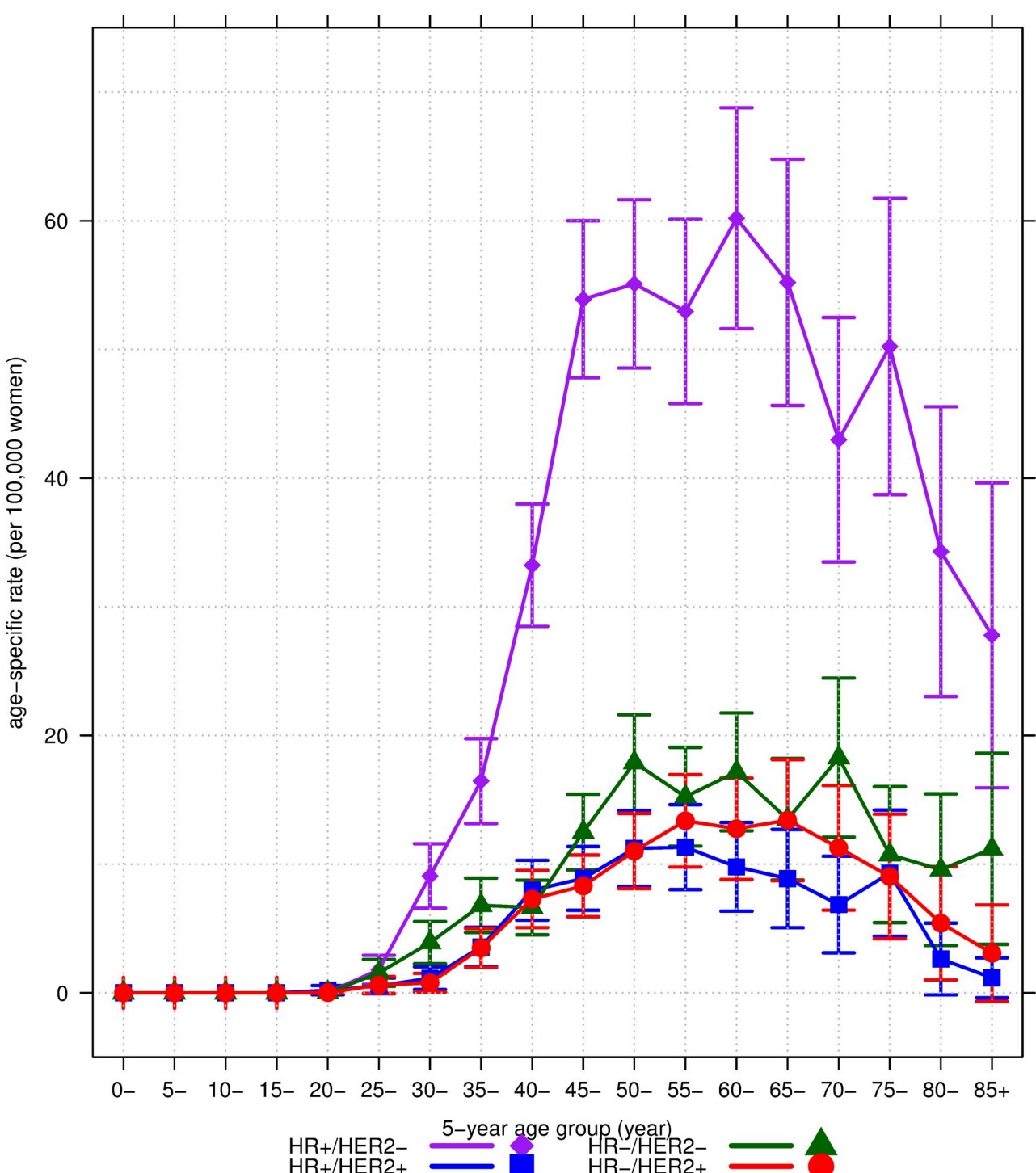

**Fig 1. The 5-year mean age-specific incidence rates and 95% confidence interval per 100,000 women of the four breast cancer subtypes according to M-IHC.** HR = hormone receptor; HER2 = human epidermal growth factor receptor 2; HR+/HER2- = luminal A-like; HR+/HER2+ = luminal B-like; HR-/HER2- = triple-negative; HR-/HER2+ = HER2-enriched.

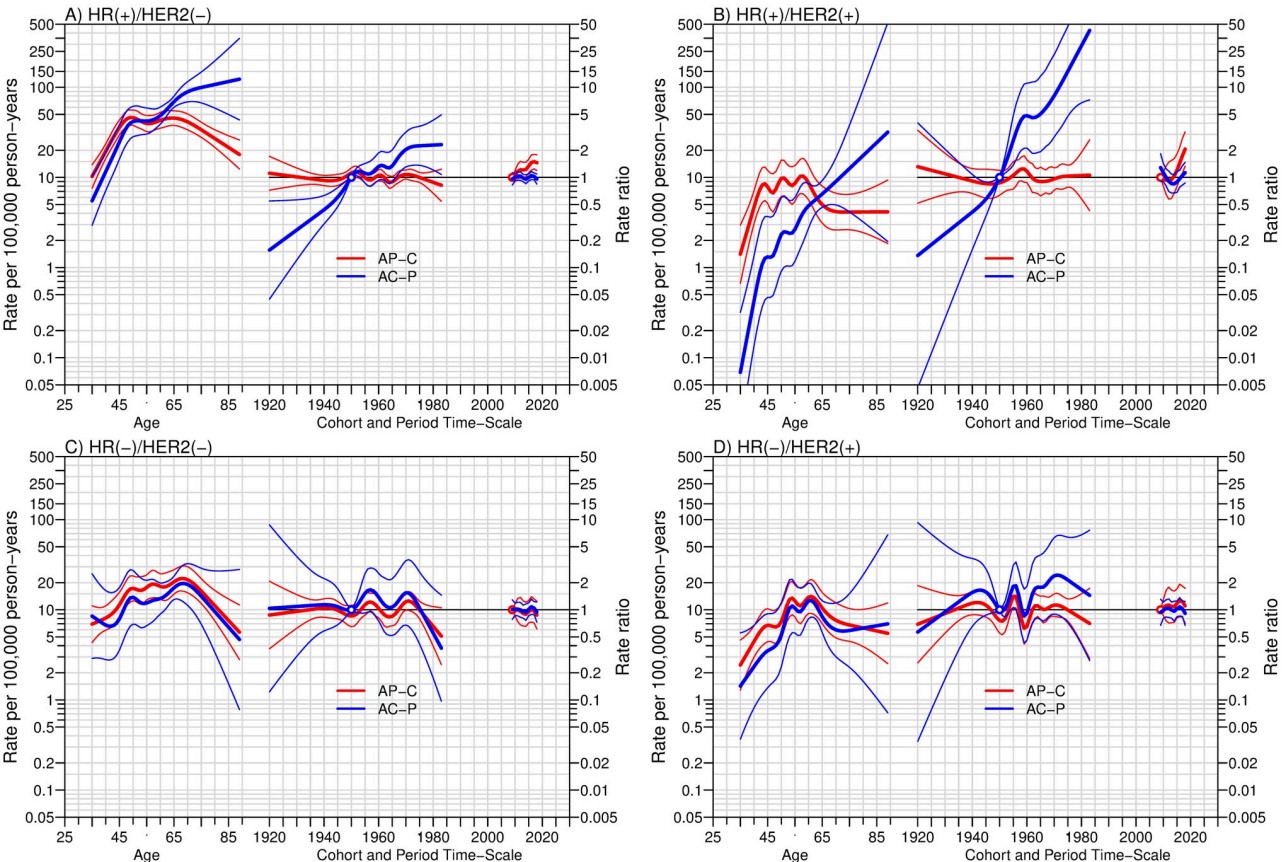

**Fig 2. The age-period-cohort (APC) trend analysis of the four breast cancer subtypes according to M-IHC in A, B, C, and D.** The three curves in each subfigure, from left to right, represent the incidence rate by age, the rate ratio of incidence by birth-cohort, and year of diagnosis for the reference cohort in blue or reference period in red. The references were the cohort born in 1950 and the diagnosis period of 2009, respectively. Thick lines and the associated thin lines are the three coefficients mentioned previously and 95% confidence intervals. AP-C = Cohort effects modeled with Age-Period component as offset; AC-P = Period effects modeled with Age-Cohort component as offset; HR = hormone receptor; HER2 = human epidermal growth factor receptor 2.

cohorts had a higher rate ratio than the older birth cohort (reference year: 1950). Near the end of the modeling spectrum, the AP-C models exhibited an increase in the rate ratio of the HR +/HER2− subtype since 2014, while the rate ratio of the HR+/HER2+ subtype later increased in 2017 (reference year: 2009). By contrast, the other two subtypes, HR−/HER2− (triple-negative) and HR−/HER2+ (HER2-enriched), were not affected by the birth cohort and year of diagnosis.

## Trends and projections of the age-standardized incidence rates

The HR+/HER2− (luminal A-like) subtype had the highest ASR of breast cancer (Fig 3). Overall, the rate of HR+/HER2− subtypes in each year was twice as high as that of the other subtypes. The ASR increased from 11.8 (95%CI: 9.5 to 14.1) cases per 100,000 women in 2009 to 20.8 (95%CI: 18.0 to 23.6) in 2018. From the joinpoint model, the incidence trends of the HR +/HER2− subtype significantly increased throughout the study period, with an annual percent change of 5.4% (95%CI: 2.5 to 8.3%). As a result, the ASR projection from the joinpoint model was 30.0 cases per 100,000 women in 2030 (Fig 3A), while the APC model projected a lower rate of 29.2 cases (Fig 3B).

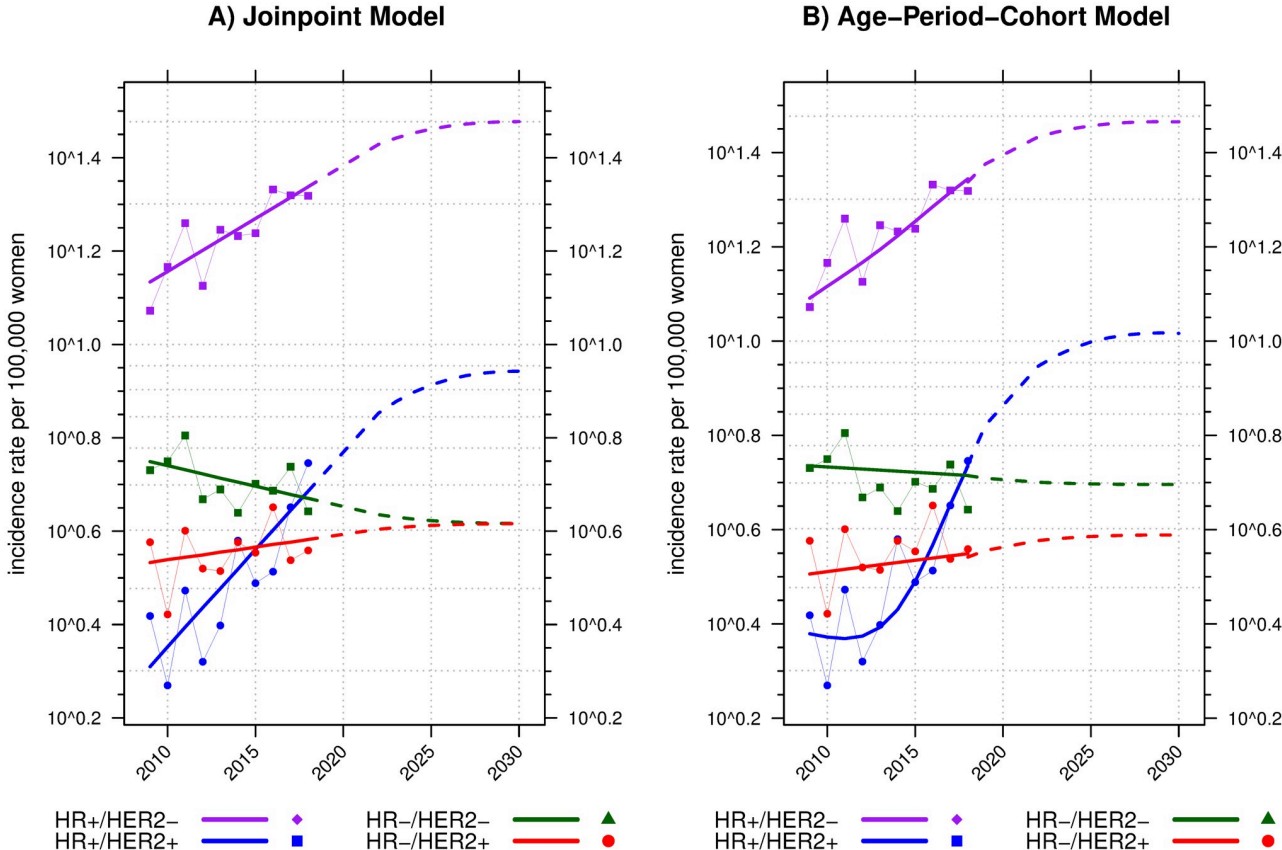

**Fig 3. The trends and projection of age-standardized rates per 100,000 women with two methods, joinpoint and age-period-cohort models in A and B of the four breast cancer subtypes according to M-IHC in four different colors.** The incidence rate (y-axis) is on a log scale. Dots connected with thin solid lines in subfigures A and B represent the calculated ASRs from 2009 to 2018. Thick solid lines represent the smooth/modeled ASRs. The extended dashed lines represent the projected trends of ASRs in 2019–2030. HR = hormone receptor; HER2 = human epidermal growth factor receptor 2.

Throughout the study period, the ASRs of other subtypes were not greater than 5.5 cases per 100,000 women. However, the rising trends in the incidence of HR+/HER2+ (luminal-B) breast cancer were remarkable. The HR+/HER2+ subtype surged from being the least to the second most common subtype, with the highest annual percent change rate of 10.1% (95%CI: 4.9 to 15.5%). The ASR of this subtype in 2009 was 2.6 per 100,000 women. It was projected to increase to 8.8 and 10.4 cases per 100,000 women in 2030 by the joinpoint and the APC models, respectively.

By contrast, the ASRs of HR–/HER2+ (HER2-enriched) and HR–/HER2– (triple-negative) subtypes were rather stable over time. The annual percent changes of the two subtypes were 1.3% (95%CI: –2.2% to 4.9%) and –2% (95%CI: –4.7% to 0.8%), respectively.

## Additional cost projection of trastuzumab treatment

The overall ASR of HER2-positive (HR+/HER2+ and HR–/HER2+) breast cancer increased from 10–11 in 2020 to 13–14 per 100,000 women in 2030 (Fig 3). This would lead to an increase in the national healthcare budget for breast cancer treatment as a part of the adjuvant therapy recommended for HER2-positive patients [9, 10]. We postulated by applying our projected rates of the HER2-positive subtype to the 2030 Thailand population projection [17].

Our forecast expected an increase of 10.8% in HER2-positive cases per year. The number of cases would reach 5,020 patients in 2020 and 10,450 in 2030. Our imputed data showed that the local and regional stages accounted for 68% of all HER2-positive cases. Thus, the expected HER2-positive eligible patients for the NLEM indication in 2030 would be 7,100. The additional cost of trastuzumab treatment was calculated under the condition that all who need the drug are treated, using the cost information based on a study sponsored by the Health Intervention and Technology Assessment Program [11]. The result illustrated an incremental cost of at least 99.4 million USD (1 USD = 34 THB).

## Discussion

In our population-based, cross-sectional study from 2009 to 2018, the most common tumor was the HR+/HER2− (luminal A-like) subtype; meanwhile, the HR+/HER2+ (luminal B-like) subtype was the less frequent subtype in the early years but became the second most common subtype at the end of the study period. The incidence of all subtypes except HR−/HER2+ (HER2-enriched) peaked at the age of 50 years, while the latter peaked at 60 years. The rates of all subtypes declined significantly after the age of 70 years.

The trends in the ASR of breast cancer were distinct according to the HR status. The incidence of HR-positive subtypes (HR+/HER2− and HR+/HER2+) increased over time. The most significant change was observed in those with HR+/HER2+ cancer. As shown in Fig 2A and 2B, the trends of HR-positive subtypes were affected by both birth cohort and period of diagnosis. The effect of period on the two subtypes increased beyond null (1) at the end of the study period in the AP-C models. The same finding was observed in the cohort effects in both subtypes. Such co-existence of both results in the same population would suggest an increased risk of being diagnosed in younger generations and later years.

Thailand has promoted breast cancer screening programs since 2002, covering over 90% of the Thai population [3]. An analysis of six provincial cancer registries in Thailand reported the effect of introducing a country-wide self-breast examination screening program [3]. It increased the detection of breast cancer among individual aged >80 years. However, the detectable rate declined several years after that because few prevalent cases were reported. Our study also found a decline in the age-specific incidence rates of all breast cancer subtypes after the age of 70 years, which is consistent with the report of a previous study.

Reproductive factors have differential effects on the trends of HR-positive breast cancer. An increase in the risk of HR-positive breast cancer was reported among women with an early age at menarche, a long period between menarche and the first delivery, fewer children, and older age at menopause, while the effect of breastfeeding history on HR-positive cancer remains controversial [25–27]. However, the 6-month exclusive breastfeeding rate in Thailand has changed over time. A series of national surveys reported that the rate was 14.5% in 2005 and increased to 23.1% in 2016, which was the highest; however, it sharply decreased to 14.0% in 2019 [28].

The reproductive characteristics of Thai women have been changing due to exposure to higher endogenous estrogen, such as younger age at menarche [29], later age at the first marriage [30], and a fall in the total fertility rate [31] from 4.9 in 1974–1976 to a fertility replacement rate of 2.1 in 1990 and down to 1.6 in 2011. These phenomena may explain the rise in the incidence of HR-positive breast cancer. Thai adolescents experienced a social dynamism in reproductive behavior, which possibly interfered with the overall change in internal estrogen exposure in this age group. From 2000 to 2012, the rate of teenage pregnancy has been increasing in Thai society [32], while the abortion rate due to unwanted pregnancies is high [33]. Such social effects on the reproductive age might have contributed to the incidence of HR-positive breast cancer.

Exogenous hormone exposure, particularly combined estrogen/progesterone hormone replacement therapy, may have also positively increased the risk of HR-positive subtypes [34, 35]. Two studies on female hormone use in Thailand demonstrated no association between exogenous hormonal use and breast cancer risk. However, these two studies did not specify the HR subtype of breast cancer [36, 37].

Elevated body mass index (BMI) is also associated with an increased risk of HR-positive breast cancer, especially in postmenopausal women [26, 38, 39]. The national surveys reported that the prevalence of overweight and obesity (BMI $\geq$ 25 mg/m$^2$) among Thai women has increased [40–42]. The prevalence increased from 25.1 in 1994 to 34.4 and 41.8 kg/m$^2$ in 2004 and 2014, respectively. In comparison, the prevalence trends of overweight and obesity in the Southern Thai women were 25.4, 36.3, and 43.7 kg/m$^2$ in the same three years, respectively. Thus, the increasing trends of BMI in southern Thai women seemed to be higher than the average BMI of Thai women. This may partly explain the rising trends in HR-positive breast cancer in Thailand.

The increase in HR-positive subtypes, particularly in the young birth cohorts and recent periods, was also evident in Malaysian women [43] and Western populations [44–49]. The worldwide changes in reproductive factors demonstrated in our discussion, and the increased BMI in many countries are the possible underlying cause of the changes in the trends in HR-positive subtypes; however, the etiologic pathways are not well understood.

The change in the ASR percentage of breast cancer by M-IHC classification impacts human resources and medical infrastructure reallocation. For example, approximately 700 new cancer cases per medical oncologist (MO) were reported in Thailand in 2020 [50, 51], while the ratio in the United States and European countries was around 100–300 cases per MO since 2015 [52]. Therefore, to meet the ratio of fewer than 300 cases per MO, additional 365 MOs are needed in Thailand; however, as of 2020 only 268 MOs were available. The number of MOs also depends on the differences in sociocultural cost and health and medical care infrastructure in each country. Such differences imply a disparity in the healthcare system but not an inequity in people's rights to healthcare access.

Our study has some limitations. First, there was a high level of missingness in the data on ER, PR, and HER2 status. The occurrence of missingness was primarily due to the non-intention to undergo further chemotherapy and radiotherapy; therefore, the degree of missingness had no backward correlation with the degree of positivity of the receptor status. We imputed missing data assuming missingness was at random and we properly modeled missing values. The missingness of receptor status was found in patients who underwent surgery at other hospitals and were referred to receive adjuvant therapy in Songklanagarind Hospital. Again, the missing data seemed to be random and not biased toward the risk status of treatable subtype. Second, inferring the results to other provinces in Southern Thailand seems possible because of the population demographics. However, the inference of this study to women in different regions of Thailand could be affected by the difference in the genetic background and associated risk factors of breast cancer that might be slightly non-homogeneous throughout the country.

In conclusion, the incidence trends of HR-positive and HER2-positive breast cancer in Thailand have been increasing, particularly in young birth cohorts and recent periods of diagnosis. The rising trends in breast cancer incidence direct the future health care budget, human resources, and medical care facilities. Their shortage in the field of cancer affects patient care quality and may influence the prognosis and survival, as well as the disparity and equity in healthcare access. Therefore, healthcare providers and administrators should prepare an appropriate plan to anticipate the situation of breast cancer shortly forecasted in this study.

## Supporting information

**S1 Table. Demographics and tumor characteristics of observed data stratified by receptor status.**
(DOCX)

**S2 Table. Demographics and tumor characteristics stratified by receptor status after imputation of unknown receptor status.**
(DOCX)

**S1 Appendix. The calculated margin of error of imputed datasets by three methods.**
(DOCX)

## Acknowledgments

The researchers profoundly thank the Songkhla population-based cancer registry and medical records of Songklanagarind Hospital, Songkhla province, Thailand, for their data provided in this analysis.

The authors would like to extend our sincere thanks to the Department of Physical Therapy staff, Faculty of Medicine, Prince of Songkla University for providing the computer, convenient place, and support during our data collection. Special thanks to Ms.Paradee Prechawittayakul, who help us to extracted the data from the cancer registry database. We also thank Ms.Thanatta Nuntadusit for contacting the other departments and Mr.Surichai Bilheem for resolving the statistic problems.

We would like to thank Editage (www.editage.com) for English language editing.

## Author Contributions

**Conceptualization:** Aungkana Chuaychai, Hutcha Sriplung.

**Data curation:** Aungkana Chuaychai.

**Formal analysis:** Aungkana Chuaychai.

**Investigation:** Aungkana Chuaychai.

**Methodology:** Aungkana Chuaychai.

**Project administration:** Aungkana Chuaychai.

**Supervision:** Hutcha Sriplung.

**Validation:** Hutcha Sriplung.

**Visualization:** Aungkana Chuaychai.

**Writing – original draft:** Aungkana Chuaychai.

**Writing – review & editing:** Aungkana Chuaychai, Hutcha Sriplung.

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
