## [Decision Letter · Decision Letter 0]

6 Oct 2021

PONE-D-21-21989A rapid rise in hormone
receptor-positive and HER2-positive breast cancer subtypes in Southern Thai women: a
population-based study in Songkhla.PLOS ONE

Dear Dr. Sriplung,

Thank you for submitting your manuscript to PLOS ONE. After careful consideration, we
feel that it has merit but does not fully meet PLOS ONE’s publication
criteria as it currently stands. Therefore, we invite you to submit a revised
version of the manuscript that addresses the points raised during the review
process. We
encourage you to make changes requested by the Reviewer and myself (see below). To
be suitable for publication, I strongly advise to seek for help from an
English speaker to review your manuscript as there are several unclear
parts. 

Please submit your revised manuscript by Nov 20 2021 11:59PM. If you will need more
time than this to complete your revisions, please reply to this message or contact
the journal office at plosone@plos.org.
When you're ready to submit your revision, log on to https://www.editorialmanager.com/pone/ and select
the 'Submissions Needing Revision' folder to locate your manuscript
file.

Please include the following items when submitting your revised
manuscript:A rebuttal letter that responds to each point raised by the academic
editor and reviewer(s). You should upload this letter as a separate file
labeled 'Response to Reviewers'.A marked-up copy of your manuscript that highlights changes made to the
original version. You should upload this as a separate file labeled
'Revised Manuscript with Track Changes'.An unmarked version of your revised paper without tracked changes. You
should upload this as a separate file labeled 'Manuscript'.

If you would like to make changes to your financial disclosure, please include your
updated statement in your cover letter. Guidelines for resubmitting your figure
files are available below the reviewer comments at the end of this letter.

We look forward to receiving your revised manuscript.

Kind regards,

Sophie Pilleron, PhD

Academic Editor

PLOS ONE

2. In your ethics statement in the Methods section and in the online submission form,
please provide additional information about the data used in your retrospective
study. Specifically, please ensure that you have discussed whether all data were
fully anonymized before you accessed them and/or whether the IRB or ethics committee
waived the requirement for informed consent. If patients provided informed written
consent to have data from their medical records used in research, please include
this information.

3. Thank you for providing the date(s) when patient medical information was initially
recorded. Please also include the date(s) on which your research team accessed the
databases/records to obtain the retrospective data used in your study.

Additional Editor Comments (if provided):

Thank you for submitting your manuscript to Plos One. The topic is of high importance
and the method seem adequate.

However I have some comments in addition to those of the reviewer.

Major points:

I would suggest authors to add more details about the cancer registry and
how a cancer case is defined. Could you tell be more about the
population covered by the registry? Does it cover all the province? The
authors mentioned that the registry reports « both
population-based and hospital-based statistics ». This is
unclear. How do a cancer case is defined? What is the date of incidence
considered? How did authors consider cancer diagnosis based on death
certificate only (DCO)?In this analysis, did authors apply an age criteria for inclusion?Also, I really like the discussion part about number of new cancer cases
and estimated costs (lines 335-344). This would deserve to be part of
the results part and to be presented as a secondary objective. This is
an interesting point that may be of interest for policy makers. I would
highlight that point. If authors decide to do so, they need to explain
about they estimate the nb of new cancer cases in 2030 and how they
estimate the excess costs in the method section.The paragraph at lines 345-353 is very interesting. Why did not the authors use the Rubin’s rules to combine their
estimates?Because authors studied trends in cancer incidence, I would like to see a
comment in the discussion on the possible change (or not) of recording
system of new cancer cases. Also, authors showed that there is a
decrease in incidence from the age of 70. Would not it be explained by a
lower ascertainment of cancer cases at older ages? Older adults may be
less likely to go through cancer diagnosis work-ups because of their
age, the presence of other conditions, or frailty. This should be
mentioned as possible reason. Another possible reason for changes in incidence is breastfeeding
practices? Have they changed over the period covered by the study?

Minor points: 

 I would suggest to replace 10e5 by 100,000 as it is easier to
read.I strongly encourage authors to have an English-native speaker to check the
grammar and sentence structure as there are several parts of text that are
unclear. For instance, lines 223-231, 243, 259, 264-266,  196-299,
331-333, 355-359, 363. I encourage authors to reformulate.The introduction may be shortened, in particular, the paragraph on racial and
cultural disparities. Because the study is about incidence, the paragraph on
treatment does not seem relevant in the introduction.Authors mentioned cancer stage several times but they did not define which
staging system they used. I would then suggest to add the definition in the
method section. Lines 304-305: is not tautological?Lines 311-312: authors mentioned an increase in teenage pregnancy. Would not
it explain by a better reporting? Lines 315-317: I would invite caution with this interpretation.Lines 317: Are not there any data at all?Lines 320-323: Could authors include some figures in the text? What are the
prevalences?Figures: I would recommend to add confidence intervals around
curves. Authors used both relative risk and rate ratios. I advise authors to pick one
and stick with it throughout the manuscript. However, relative risk and rate
ratios are not similar. Authors should probably clarify which one is
estimated by APC models.Line 109: what means stage 0 or extent 1?Line 118: authors wrote « [the registry] collects cancer patients
diagnosed or treated ». This is unclear. How does the cancer registry
collect data on new cancer cases?Line 161: Which period did authors consider as
« recent »?

Reviewers' comments:

Reviewer's Responses to Questions

**Comments to the Author**

1. Is the manuscript technically sound, and do the data support the conclusions?

Reviewer #1: Yes

2. Has the statistical analysis been performed
appropriately and rigorously? 

Reviewer #1: Yes

3. Have the authors made all data underlying the
findings in their manuscript fully available?

Reviewer #1: Yes

4. Is the manuscript presented in an intelligible
fashion and written in standard English?

Reviewer #1: Yes

5. Review Comments to the Author

Reviewer #1: The authors present the first population-based cancer registry
data to summarize molecular-immune-histochemical subtypes. Overall, the authors
provide a very rigorous analysis to interpret trends in hormone receptor status of
primary breast cancer cases from 2009 to 2018. This is an important research topic,
but I have some concerns, mainly related to the level of missing in the data.

Introduction

I would suggest a little reorganizing of this paragraph “Since 2002, Thailand
has implemented the universal health coverage (UHC) scheme [10]. It is the basic
scheme for Thai citizens, covered health promotion, disease screening, treatment of
basic and high-cost diseases under the Thai national list of essential medicines 5
(NLEM), and palliative/supportive care. The UHC scheme contributed approximately 79%
of the Thai population’s health insurance [11].”. Mainly, I would
recommend mentioning that 79% of the population’s health insurance is the
basic covered should be mentioned after the fact it is the basic scheme for Thai
citizens. I would also recommend switching covered to covers as the UHC appears to
still be in place.

The authors stated “Though a targeted therapy is cost-effective among early
breast cancer cases, it increases the burden of cancer care costs as a whole and
significantly contributes to the national health budget, especially when early
patients are more detected“. I am not exactly following your logic on how it
would significantly increase the burden of cancer care costs as a whole if it is a
cost-effective treatment. Can you please elaborate?

The authors state “Other targeted therapeutic drugs for breast cancer have
been or are in Thai FDA”. Can you please provide a few examples?

Methods

The authors mention that they excluded DCIS cases. Yet, DCIS can also be receptor
positive. Can the authors explain why this did not include this group?

Management of missing data

The authors report using multiple imputation to account for missing outcome data, but
more information would be valuable. Specifically, you mention that the missing in
other variables is 22%, but what is the missing for your receptor statuses? This
seems like a much bigger concern. It is also important to know how much change this
imputation process had on your findings. Mainly, what does the results look like in
the non-imputed dataset? How different is the population?

I am also a little confused by the process. Did you enter the other missing variables
as missing or did you use the imputed predictor variables to determine?

Results

Can you please include the range of age at breast cancer diagnosis in addition to the
IQR?

It would also be good to see some comparison of demographics by HR status.

From reading the results section, it appears that almost 50% of the receptor status
was missing. This is of great concern, especially given that there was an additional
22% of other missing variables. There really needs to be comparisons in the outcome
and characteristics of those with and without receptor status, and the new sample
with the imputed data to understand the impact.

It is great to see that you included observed data versus imputed data in the results
section. I would ask that you include percent in the column “Number of
cases”. I still think there needs to be a comparison of how the demographics
of the women looked before and after imputation.

Discussion

The sentence “In contrast, reproductive factors have an inconclusive influence
on HR-negative cancers.” Requires a reference

6. PLOS authors have the option to publish the peer
review history of their article (what does this mean?). If
published, this will include your full peer review and any attached files.

If you choose “no”, your identity will remain anonymous but your review
may still be made public.

**Do you want your identity to be public for this peer review?** For
information about this choice, including consent withdrawal, please see our
Privacy Policy.

Reviewer #1: No

---

## [Author Response · Author response to Decision Letter 0]

23 Dec 2021

RESPONSE TO EDITOR AND REVIEWER

RESPONSE TO THE EDITOR:

MAJOR POINTS:

COMMENT: I would suggest authors to add more details about the cancer registry and
how a cancer case is defined. Could you tell be more about the population covered by
the registry? Does it cover all the province? The authors mentioned that the
registry reports « both population-based and hospital-based statistics
». This is unclear. How do a cancer case is defined? What is the date of
incidence considered? How did authors consider cancer diagnosis based on death
certificate only (DCO)?

RESPONSE: We added more detail about the Songkhla cancer registry and revised the
unclear sentences (Study setting and data sources paragraph: 3-4).

“The Songkhla Cancer Registry is a population-based cancer registry that
enrolled cancer patients residing in Songkhla Province. Case ascertainment began in
1988, and patients’ data were regularly supplied to the Cancer Incidence in
Five Continents (CIV since vol. VIII) [15]. The cases were identified using active
and passive case-finding methods and registered by trained staff. The sources of
information included medical records and pathological records of all hospitals in
the province. The status of the patients in the database is reported when deaths
occur in the hospitals and healthcare network and is confirmed by the Bureau of
Registration Administration Database, Department of Provincial Administration,
Ministry of Interior.

 The diagnosis date of cancer patients was obtained from the most reliable source, if
possible. The diagnosis date was indicated as the date of biopsy, the date when the
specimen was received at the pathological laboratory, or the report date. When the
date of definitive diagnosis provided by the pathological laboratory was not
indicated, the date of hospital admission due to this malignancy and other dates
related to the occurrence of malignancy or death were considered [16].” 

COMMENT: In this analysis, did authors apply an age criteria for inclusion?

RESPONSE: We included all breast cancer cases who were diagnosed in the study
periods. However, from our data, the youngest patient who met our criteria was 24
years old. That is why we set the age group for calculating the age-specific
incidence from age 20 to ≥85 years.

COMMENT: Also, I really like the discussion part about number of new cancer cases and
estimated costs (lines 335-344). This would deserve to be part of the results part
and to be presented as a secondary objective. This is an interesting point that may
be of interest for policy makers. I would highlight that point. If authors decide to
do so, they need to explain about they estimate the nb of new cancer cases in 2030
and how they estimate the excess costs in the method section.

RESPONSE: We decided to set the projection of additional cost due to the trastuzumab
treatment in 2030 as the secondary objective of our study. We explained how to
estimate the excess cost in the methodology part as subtopic “Trastuzumab
cost projection” and added the outcome in the result part under the subtopic
“Additional cost projection of trastuzumab treatment.”

COMMENT: Why did not the authors use the Rubin's rules to combine their
estimates?

RESPONSE: In our analysis, we calculated the estimates and margin of error of the
imputed data by three methods, median and its 95% probability interval, mean, and
Rubin’s estimate of the 95% confidence interval and mean ±2 SD
margins. All three methods gave very close values of the estimates and the margins
of error. Thus, in the previous draft, we decided to show only the median proportion
and 95%PI, but we forgot to explain the similarity of the results. So, in this
revision, we showed the results from all methods in Table 2. (Table 2 and Results
part in subtopic: Proportions of breast cancer M-IHC subtypes)

COMMENT: Because authors studied trends in cancer incidence, I would like to see a
comment in the discussion on the possible change (or not) of recording system of new
cancer cases. Also, authors showed that there is a decrease in incidence from the
age of 70. Would not it be explained by a lower ascertainment of cancer cases at
older ages? Older adults may be less likely to go through cancer diagnosis workups
because of their age, the presence of other conditions, or frailty. This should be
mentioned as possible reason.

RESPONSE: One study from six provincial cancer registries in Thailand found an effect
of introducing a country-wide self-breast examination screening program under the
universal health insurance, which covered over 90% of the Thai population increase
capturing of the hidden breast cancer cases aged over 80. The capture rate declined
after some years of the program's start even it has continued as many cases have
been captured. So, we think that we captured the decline in the ascertainment rate
during our study period and is becoming stabilized after the end of our study. In
the paragraph, we discussed that “Thailand has promoted breast cancer
screening programs since 2002, covering over 90% of the Thai population [3]. An
analysis of six provincial cancer registries in Thailand reported the effect of
introducing a country-wide self-breast examination screening program [3]. It
increased the detection of breast cancer among individual aged >80 years.
However, the detectable rate declined several years after that because few prevalent
cases were reported. Our study also found a decline in the age-specific incidence
rates of all breast cancer subtypes after the age of 70 years, which is consistent
with the report of a previous study.” (Discussion: paragraph 3).

COMMENT: Another possible reason for changes in incidence is breastfeeding practices?
Have they changed over the period covered by the study?

RESPONSE: We added the detail about the change in the breastfeeding rate in Thailand
over time in the revised version (Discussion: paragraph 4).

“However, the 6-month exclusive breastfeeding rate in Thailand has changed
over time. A series of national surveys reported that the rate was 14.5% in 2005 and
increased to 23.1% in 2016, which was the highest; however, it sharply decreased to
14.0% in 2019 [27].”

27. Topothai C, Tangcharoensathien V. Achieving global targets on breastfeeding in
Thailand: gap analysis and solutions. Int Breastfeed J. 2021 Dec;16(1):38. doi:
10.1186/s13006-021-00386-0. PMID: 33962645.

MINOR POINTS:

COMMENT: I would suggest to replace 10e5 by 100,000 as it is easier to read.

RESPONSE: We agree with the editor's comment and change text "105" to "100,000" in
our revised version.

COMMENT: I strongly encourage authors to have an English-native speaker to check the
grammar and sentence structure as there are several parts of text that are unclear.
For instance, lines 223-231, 243, 259, 264-266, 196-299, 331-333, 355-359, 363. I
encourage authors to reformulate.

RESPONSE: Before resubmission, we sent the revised manuscript to the Editage for
checking the grammar and sentence structure. 

COMMENT: The introduction may be shortened, in particular, the paragraph on racial
and cultural disparities. Because the study is about incidence, the paragraph on
treatment does not seem relevant in the introduction.

RESPONSE: We decided to delete the paragraph that describes the racial variations in
Thailand. However, we remained the paragraph that described the treatment option in
breast cancer patients. It shows why the knowledge of breast cancer incidence in
terms of subtype is essential. In addition, it relates to our secondary objective
that we added in the revised version. 

COMMENT: Authors mentioned cancer stage several times but they did not define which
staging system they used. I would then suggest to add the definition in the method
section.

RESPONSE: Songkhla cancer registry records the breast cancer stage according to the
AJCC-TNM classification. In a long-running cancer registry, cancer staging may
change over time. The registries usually reclassify the extent of disease to the
summary stage according to the SEER staging system. It categorized tumor stage to in
situ, localized, regionalized, distant, and unknown stage.

COMMENT: Lines 304-305: is not tautological? 

RESPONSE: We revised the sentences in Line 304-305 and described this issue in more
detail as “However, the 6-month exclusive breastfeeding rate in Thailand has
changed over time. A series of national surveys reported that the rate was 14.5% in
2005 and increased to 23.1% in 2016, which was the highest; however, it sharply
decreased to 14.0% in 2019 [27].” (Discussion: paragraph 4).

COMMENT: Lines 311-312: authors mentioned an increase in teenage pregnancy. Would not
it explain by a better reporting?

RESPONSE: We changed the last part of the discussion on reproductive behaviors in the
Thai population to lessen the stress on teenage pregnancy and HR-positive breast
cancer. The detail is, “Thai adolescents experienced a social dynamism in
reproductive behavior, which possibly interfered with the overall change in internal
estrogen exposure in this age group. From 2000 to 2012, the rate of teenage
pregnancy has been increasing in Thai society [31], while the abortion rate due to
unwanted pregnancies is high [32]. Such social effects on the reproductive age might
have contributed to the incidence of HR-positive breast cancer.”

 In addition, we also changed reference no. 31 (previous version no.29) to
“Sukrat B. Thailand Adolescent Birth Rate: Trend and Related Indicators. Thai
J Obstet Gynaecol. 2014 Jan 20;15–21”. (Discussion: paragraph 5)

COMMENT: Lines 315-317: I would invite caution with this interpretation.

RESPONSE: We revised the sentences, described them in more detail. The new text is
“Exogenous hormone exposure, particularly combined estrogen/progesterone
hormone replacement therapy, may have also positively increased the risk of
HR-positive subtypes [33,34]. Two studies on female hormone use in Thailand
demonstrated no association between exogenous hormonal use and breast cancer risk.
However, these two studies did not specify the HR subtype of breast cancer
[35-36].”

 In addition, we changed the reference to “Type and timing of menopausal
hormone therapy and breast cancer risk: individual participant meta-analysis of the
worldwide epidemiological evidence. The Lancet. 2019
Sep;394(10204):1159–68”. (Discussion: paragraph 6)

COMMENT: Lines 317: Are not there any data at all?

RESPONSE: Of course. We found only a few studies [35,36] on female hormone use and
breast cancer risk. However, they are not specific on breast cancer subtypes or
receptor status. They did not find the association between hormone use and breast
cancer risk. We described more in the revised version. (Discussion: paragraph 6)

35. Poosari A, Promthet S, Kamsa-ard S, Suwanrungruang K, Longkul J, Wiangnon S.
Hormonal Contraceptive Use and Breast Cancer in Thai Women. J Epidemiol. 2014 May
5;24(3):216–20. 

36. Ratanawichitrasin A, Bhodhisuwan K, Reansuwan W, Kongpatanakul S,
Ratanawichitrasin S. Risk of Breast Cancer in Post-Menopausal Women Using Hormone
Replacement Therapy. J Med Assoc Thai. 2002 May;85(5):583-9.

COMMENT: Lines 320-323: Could authors include some figures in the text? What are the
prevalences?

RESPONSE: We added more detail to the paragraph that described the BMI and
positive-breast cancer risk and prevalence of obesity in Thai women as “The
national surveys reported that the prevalence of overweight and obesity (BMI
≥ 25 mg/m2) among Thai women has increased [39-41]. The prevalence increased
from 25.1 in 1994 to 34.4 and 41.8 kg/m2 in 2004 and 2014, respectively. In
comparison, the prevalence trends of overweight and obesity in the Southern Thai
women were 25.4, 36.3, and 43.7 kg/m2 in the same three years, respectively. Thus,
the increasing trends of BMI in southern Thai women seemed to be higher than the
average BMI of Thai women. This may partly explain the rising trends in HR-positive
breast cancer in Thailand.” (Discussion: paragraph 7).

COMMENT: Figures: I would recommend to add confidence intervals around curves.

RESPONSE: We added the 95% confidence interval in Fig 1 following the editor’s
suggestion and change y-scale from log- to normal scale.

COMMENT: Authors used both relative risk and rate ratios. I advise authors to pick
one and stick with it throughout the manuscript. However, relative risk and rate
ratios are not similar. Authors should probably clarify which one is estimated by
APC models.

RESPONSE: We already clarified that the RR is the rate ratio like we notice in figure
2. We already revised the error text in our manuscript. 

COMMENT: Line 109: what means stage 0 or extent 1?

Line 118: authors wrote « [the registry] collects cancer patients diagnosed or
treated ». This is unclear. How does the cancer registry collect data on new
cancer cases?

Line 161: Which period did authors consider as « recent »?

RESPONSE: Line 109: We included only invasive tumors, added “invasive”
(subtopic: Study design and participants, paragraph 2) So, we can delete “in
situ tumor (code as stage 0 or extent 1)”.

Line 118: We already added the detail about the cancer registry. (Setting and data
sources: paragraph 3-4).

Line 161: we specified the period when we used the coefficients to project the future
trend “coefficients from 2009-2018”. (Subtopic: Incidence rates,
trends, and projection, paragraph 4)

RESPONSE TO REVIEWER 1:

COMMENT: The authors present the first population-based cancer registry data to
summarize molecular-immune-histochemical subtypes. Overall, the authors provide a
very rigorous analysis to interpret trends in hormone receptor status of primary
breast cancer cases from 2009 to 2018. This is an important research topic, but I
have some concerns, mainly related to the level of missing in the data.

RESPONSE: Our analysis found high percentages of missing data on ER, PR, and HER2
status, 45.6, 45.6, and 49.8%, respectively. Literature reviews found that multiple
imputation could reduce bias and improved effect estimation at the high proportion
of missing data. It is especially when imputation was conducted based on the
auxiliary information, including variables predicting the complete set of the
hormone and HER2 receptor statuses in the model. Thus, the replacement of the cases
with missing values of the ER, PR, and HER2 statuses was random and conformed to the
original construct of the association of the non-missing cases. The maximum of
missing proportions that can hold by the multiple imputation method is different.
However, the evidence shows that the multiple imputation by chained equation (MICE)
method can handle up to 80% of missingness [3].

References.

1. Eisemann N, Waldmann A, Katalinic A. Imputation of missing values of tumour stage
in population-based cancer registration. BMC Med Res Methodol. 2011 Dec;11(1):129. 

2. Madley-Dowd P, Hughes R, Tilling K, Heron J. The proportion of missing data should
not be used to guide decisions on multiple imputation. J Clin Epidemiol. 2019
Jun;110:63–73. 

3. Souverein OW, Zwinderman AH, Tanck MWT. Multiple Imputation of Missing Genotype
Data for Unrelated Individuals. Annals of Human Genetics. 2006;70(3):372–81. 

COMMENT: 

Introduction

I would suggest a little reorganizing of this paragraph "Since 2002, Thailand has
implemented the universal health coverage (UHC) scheme [10]. It is the basic scheme
for Thai citizens, covered health promotion, disease screening, treatment of basic
and high-cost diseases under the Thai national list of essential medicines 5 (NLEM),
and palliative/supportive care. The UHC scheme contributed approximately 79% of the
Thai population's health insurance [11].". 

Mainly, I would recommend mentioning that 79% of the population's health insurance is
the basic covered should be mentioned after the fact it is the basic scheme for Thai
citizens. I would also recommend switching covered to covers as the UHC appears to
still be in place.

RESPONSE: We revised the sentence sequencing in the paragraph, “Since 2002,
Thailand has implemented a universal health coverage (UHC) scheme [7]. This is the
primary insurance scheme for Thai citizens. As of 2020, the UHC scheme has already
provided health services to approximately 79% of the Thai population [8]. It covers
health promotion, disease screening, treatment of basic and high-cost diseases under
the Thai National List of Essential Medicines (NLEM), and palliative/supportive
care.” following the suggestion from the reviewer. (Introduction: paragraph
3)

COMMENT: 

The authors stated "Though a targeted therapy is cost-effective among early breast
cancer cases, it increases the burden of cancer care costs as a whole and
significantly contributes to the national health budget, especially when early
patients are more detected ". 

I am not exactly following your logic on how it would significantly increase the
burden of cancer care costs as a whole if it is a cost-effective treatment. Can you
please elaborate?

The authors state "Other targeted therapeutic drugs for breast cancer have been or
are in Thai FDA". Can you please provide a few examples?

RESPONSE: 

 Though a targeted therapy is cost-effective among early breast cancer cases, it
increases the sum of cancer care budget by 15560 USD per one patient to add a
targeted therapy to the to the baseline treatment while the QALYs were expected to
increase by 4.59 years. While an increased burden to the national health budget
especially when early patients are more detected, in turn, it implies that the
longer life gain to women benefited from a targeted therapy would make a return in
terms of the GDP per capita greater than the budget spent by the UC scheme for the
treatment cost per patient. 

 In addition, we added information about the other targeted drug that may include in
the Thai national list of essential medicines (NLEM) in the future (Introduction:
paragraph 6). It is the reason to show that why the estimated incidence in each
breast cancer subtype is essential. The additional sentences are “Other
targeted drugs approved by the Thai Food and Drug Administration for breast cancer
treatment include pertuzumab and ribociclib. Currently, these are the only two drugs
available to patients who are covered by the civil servant medical benefit scheme
[12,13].”

COMMENT: 

Methods

The authors mention that they excluded DCIS cases. Yet, DCIS can also be
receptor-positive. Can the authors explain why this did not include this group?

RESPONSE: In our population-based cancer registry of Songkhla, it was not designed to
collect in situ cases of breast cancer from the start of the registration since the
standard of data collection for DCIS tumors is nonhomogeneous in hospitals in the
province. It is possible in the future to collect this cancer at the best
completeness after the province-wide data collection procedures are
standardized.

COMMENT: 

Management of missing data

The authors report using multiple imputation to account for missing outcome data, but
more information would be valuable. Specifically, you mention that the missing in
other variables is 22%, but what is the missing for your receptor statuses? This
seems like a much bigger concern. It is also important to know how much change this
imputation process had on your findings. Mainly, what does the results look like in
the non-imputed dataset? How different is the population? I am also a little
confused by the process. Did you enter the other missing variables as missing or did
you use the imputed predictor variables to determine?

RESPONSE: In our study, we imputed only the missing values in outcome variables,
which are ER, PR, and HER2 status. The proportion of missing values in each variable
were 45.6, 45.6, and 49.8%, respectively. For the question that which of the missing
values were addressed in the imputation, we transformed variables other than the
three receptor statuses to factors and set the missing values to an
‘unknown’ class of that variable, so that the ‘unknown’
values have a predictive ability on the receptor statuses in the imputation. We also
used the values of one of the three receptor statuses to predict the missing values
of the other missing receptor statuses. The final set of the dataset consisted of
1000 imputed data set, a part of them were the known values from the original data,
another part of them are imputed data filled into the missing (blank) values.

 In Table 1, we added the characteristics of ER, PR, and HER2 status. For example,
the numbers of HER2 negative/equivocal, positive, unknown were 1127, 319, 1437,
after imputation, the mean of 1000 imputed data set were 2214, 669, and 0, while the
SD of the imputed dataset were 38, 38 and 0. After we got the imputed values of the
three receptor statuses, the Table 2 described the mean/median of the observed and
imputed data with the range of prediction by the three methods. The difference in
the observed and imputed proportion is on the third or second decimal places. 

 In addition, in our revised manuscript, we added the comparison baseline
characteristics between observed and imputed data, as the Table 1 in the result
part, and added a new table in the supplement (S1A and S1B Table).

COMMENT: 

Results

Can you please include the range of age at breast cancer diagnosis in addition to the
IQR?

RESPONSE: The range of age was 24-95 years. We added the range of age in table1.

COMMENT: 

Results

It would also be good to see some comparison of demographics by HR status.

From reading the results section, it appears that almost 50% of the receptor status
was missing. This is of great concern, especially given that there was an additional
22% of other missing variables. There really needs to be comparisons in the outcome
and characteristics of those with and without receptor status, and the new sample
with the imputed data to understand the impact.

It is great to see that you included observed data versus imputed data in the results
section. I would ask that you include percent in the column "Number of cases". I
still think there needs to be a comparison of how the demographics of the women
looked before and after imputation.

RESPONSE: We added the percentage in column "Number of cases" of Table 2. The detail
about the missing proportion and how we imputed the missingness is described in the
previous comment. In addition, we added the baseline characteristic comparison
between observed and imputed data according to the comment in table 1 and additional
2 tables (S1A S1B Table) in the supplement. 

COMMENT: 

Discussion

The sentence "In contrast, reproductive factors have an inconclusive influence on
HR-negative cancers." Requires a reference

RESPONSE: We decided to revise this paragraph and delete the sentence according to
the editor’s comment, making it a clearer comparison. From “In
contrast, reproductive factors have an inconclusive influence on HR-negative
cancers.” to “The reproductive characteristics of Thai women have been
changing due to exposure to higher endogenous estrogen, such as younger age at
menarche [28], later age at the first marriage [29], and a fall in the total
fertility rate [30] from 4.9 in 1974–1976 to a fertility replacement rate of
2.1 in 1990 and down to 1.6 in 2011. These phenomena may explain the rise in the
incidence of HR-positive breast cancer. Thai adolescents experienced a social
dynamism in reproductive behavior, which possibly interfered with the overall change
in internal estrogen exposure in this age group. From 2000 to 2012, the rate of
teenage pregnancy has been increasing in Thai society [31], while the abortion rate
due to unwanted pregnancies is high [32]. Such social effects on the reproductive
age might have contributed to the incidence of HR-positive breast cancer.”
(Discussion: paragraph 5)

to reviewers.docx
---

## [Decision Letter · Decision Letter 1]

25 Jan 2022

PONE-D-21-21989R1A rapid rise in hormone
receptor-positive and HER2-positive breast cancer subtypes in Southern Thai women: a
population-based study in Songkhla.PLOS ONE

Dear Dr. Sriplung,

Thank you for submitting your manuscript to PLOS ONE. After careful consideration, we
feel that it has merit but does not fully meet PLOS ONE’s publication
criteria as it currently stands. Therefore, we invite you to submit a revised
version of the manuscript that addresses the minor points raised during the review
process.

Please submit your revised manuscript by Mar 11 2022 11:59PM. If you will need more
time than this to complete your revisions, please reply to this message or contact
the journal office at plosone@plos.org.
When you're ready to submit your revision, log on to https://www.editorialmanager.com/pone/ and select
the 'Submissions Needing Revision' folder to locate your manuscript
file.

Please include the following items when submitting your revised
manuscript:A rebuttal letter that responds to each point raised by the academic
editor and reviewer(s). You should upload this letter as a separate file
labeled 'Response to Reviewers'.A marked-up copy of your manuscript that highlights changes made to the
original version. You should upload this as a separate file labeled
'Revised Manuscript with Track Changes'.An unmarked version of your revised paper without tracked changes. You
should upload this as a separate file labeled 'Manuscript'.

If you would like to make changes to your financial disclosure, please include your
updated statement in your cover letter. Guidelines for resubmitting your figure
files are available below the reviewer comments at the end of this letter.

We look forward to receiving your revised manuscript.

Kind regards,

Sophie Pilleron, PhD

Academic Editor

PLOS ONE

Journal Requirements:

Additional Editor Comments:

I thank authors to have well answered my comments and those from the Reviewer.

In addition to Reviewer's comments, I would add additional very minor points:

I would advise deleting "and encourage all oncologists and health policymakers
to manage  breast cancer cases in Thailand." in line 20 as it seems not
the right place to make this call.

Line 119: April 20, 2020, October 18, 2020. —> April 20,
2020,  **to**
October 18, 2020.

Line 125: The status of the patients —> the
**vital** status of the patients

Line 128: The diagnosis date of cancer patients —> **The date of
incidence**.

Lines 129-130: Because all these dates may not be the same, please, could you be more
specific on how you choose the date? The earlier available? 

Line 137: Does it means that the cancer registry can include patients from other
province too? If so, the population covered is not only that of the Songkhla
Province. Am I right?

Line 431: what do you mean? Missing at random?

Line 435: I am not sure to understand the link between this sentence and the
following one.

Reviewers' comments:

Reviewer's Responses to Questions

**Comments to the Author**

1. If the authors have adequately addressed your comments raised in a previous round
of review and you feel that this manuscript is now acceptable for publication, you
may indicate that here to bypass the “Comments to the Author” section,
enter your conflict of interest statement in the “Confidential to
Editor” section, and submit your "Accept"
recommendation.

Reviewer #1: All comments have been addressed

2. Is the manuscript technically sound, and do the data
support the conclusions?

Reviewer #1: Yes

3. Has the statistical analysis been performed
appropriately and rigorously? 

Reviewer #1: Yes

4. Have the authors made all data underlying the
findings in their manuscript fully available?

Reviewer #1: Yes

5. Is the manuscript presented in an intelligible
fashion and written in standard English?

Reviewer #1: Yes

6. Review Comments to the Author

Reviewer #1: Overall, the authors did an excellent job addressing all of my
comments. I just had a few nitpicky things I still noticed.

Introduction:

Please add a reference to the first line “Breast cancer is the most common
cancer among women worldwide”.

Methods:

If you one method to determine malignancy is based on death, wouldn’t this
mean there may be an underestimation of the true prevalence of the disorder? I
assume not everyone who dies undergoes an autopsy. This should be addressed in the
limitations section.

You mention that your youngest case of breast cancer is in a women who is 24 years of
age, yet use 5 year age groups to calculate the age-specific rates. What is the
number of individuals who are 24 years of age? This group is likely very small, and
unstable. How does the estimate change if you collapse the 24 year old into the
upper age group?

I was a little confused by this statement “including our projection of
age-specific incidence rates of the HER2- 194 positive subtype in 18 age groups
(0–4, 5–9, 10–14, ..., 80–84, and ≥85
years)”. How is this possible if you have the first case of cancer starting
at 24 years? Wouldn’t the relative cost for each of these other groups be
zero?

Results:

Please include range of age in this sentence “The median age of the patients
at diagnosis of breast cancer was 53.0 years 237 (interquartile range, IQR:
46.0–62.0)”.

Please also extend the table for age to show each 10 year age group from 20 to give
an idea of the instability in estimates for the younger age groups.

Figure 3 title says 105, please change to 100,000. Same in the sentence “Over
the study period, the ASRs of other subtypes were not greater than 5.5 cases per 105
women” And “. The ASR of this subtype in 2009 was 2.6 per 105
women.” And here “The overall ASR of HR-positive and HER2-positive
(HR+/HER2+ and HR-/HER2+) breast cancer increase from 30-32 and 10-11 to 39 and
13-14 per 105 women in 2020 to 2030”

7. PLOS authors have the option to publish the peer
review history of their article (what does this mean?). If
published, this will include your full peer review and any attached files.

If you choose “no”, your identity will remain anonymous but your review
may still be made public.

**Do you want your identity to be public for this peer review?** For
information about this choice, including consent withdrawal, please see our
Privacy Policy.

Reviewer #1: No

---

## [Author Response · Author response to Decision Letter 1]

7 Feb 2022

Response to Reviewers

Comment for PONE-D-21-21989R1

Journal Requirements:

Suggestion: Please review your reference list to ensure that it is complete and
correct. If you have cited papers that have been retracted, please include the
rationale for doing so in the manuscript text, or remove these references and
replace them with relevant current references. Any changes to the reference list
should be mentioned in the rebuttal letter that accompanies your revised manuscript.
If you need to cite a retracted article, indicate the article’s retracted
status in the References list and also include a citation and full reference for the
retraction notice.

Response: 

In manuscript version 2 (PONE-D-21-21989R1), revised from manuscript version
1(PONE-D-21-21989):

1. We removed the references from manuscript version 1 because the previous
references were not relevant after we revised the text in the manuscript. The lists
of removed references are as:

 Hays J. People of Thailand: origin, different Thai groups and Siamese twins | facts
and details. [cited 15 Jun 2021]. Available from: http://factsanddetails.com/southeast-asia/Thailand/sub5_8c/entry-3209.html

 Srisontisuk DS, Katchamat P, Pakdee P. Poverty and the ethnic minority groups in
Thailand. J Mekong Society. 2005;1: 151–189. 

 Reid LA. Benedict’s Austro-Tai hypothesis—an evaluation. Asian
Perspect. 1984;26: 19–34.

 Aekplakorn W, Inthawong R, Kessomboon P, Sangthong R, Chariyalertsak S, Putwatana P,
et al. Prevalence and trends of obesity and association with socioeconomic status in
Thai adults: national health examination surveys, 1991–2009. J Obes.
2014;2014: 410259. doi:10.1155/2014/410259. PMID: 24757561.

 Sakboonyarat B, Pornpongsawad C, Sangkool T, Phanmanas C, Kesonphaet N,
Tangthongtawi N, et al. Trends, prevalence and associated factors of obesity among
adults in a rural community in Thailand: serial cross-sectional surveys, 2012 and
2018. BMC Public Health. 2020;20. doi:10.1186/s12889-020-09004-w. PMID:
32493314.

2. We added some references in manuscript version 2 because we added the new text in
manuscript 2. The lists of added references are as:

 Drug and Medical Supply Information Center, Ministry of Public Health. Medical
Reimbursement Criteria for Cancer and Hematology Patients Who Need Expensive Drugs
(Additional) (W 339, W 340) [Internet]. [cited 2021 Oct 29]. Available from:
http://dmsic.moph.go.th/index/detail/7823

 Drug and Medical Supply Information Center, Ministry of Public Health. Medical
Reimbursement Criteria for Cancer and Hematology Patients Who Need Expensive Drugs
(Additional) (W 278, W 279) [Internet]. [cited 2021 Oct 29]. Available from:
http://dmsic.moph.go.th/index/detail/8221

 Bray F, Colombet M, Ferlay J, Mery L, Piñeros M, Znaor A, et al. Cancer
Incidence in Five Continents Volume XI [Internet]. [cited 2021 Oct 31]. Available
from: https://publications.iarc.fr/Book-And-Report-Series/Iarc-Scientific-Publications

 European Network of Cancer Registries (ENCR).pdf [Internet]. [cited 2021 Oct 31].
Available from: https://www.encr.eu/sites/default/files/pdf/incideng.pdf

 Topothai C, Tangcharoensathien V. Achieving global targets on breastfeeding in
Thailand: gap analysis and solutions. Int Breastfeed J. 2021 Dec;16(1):38. doi:
10.1186/s13006-021-00386-0. PMID: 33962645.

 Poosari A, Promthet S, Kamsa-ard S, Suwanrungruang K, Longkul J, Wiangnon S.
Hormonal Contraceptive Use and Breast Cancer in Thai Women. J Epidemiol. 2014 May
5;24(3):216–20.

 Ratanawichitrasin A, Bhodhisuwan K, Reansuwan W, Kongpatanakul S, Ratanawichitrasin
S. Risk of Breast Cancer in Post-Menopausal Women Using Hormone Replacement Therapy.
J Med Assoc Thai. 2002 May;85(5):583-9.

 Thai National Health Examination Survey 1994-1995, NHES I [Internet]. 1st ed.
Nonthaburi: Health Systems Research Institute (HSRI); 1996 [cited 2021 Oct 31]. 271
p. Available from: https://www.hiso.or.th/hiso/picture/reportHealth/report/report5.pdf

 Porapakkham Y, Boonyaratapan P, editors. Thai National Health Examination Survey
2003-2004, NHES III [Internet]. 1st ed. Nonthaburi: Health Systems Research
Institute (HSRI); 2006 [cited 2021 Oct 31]. 267 p. Available from: https://www.hiso.or.th/hiso/picture/reportHealth/report/report2.pdf

 Aekphakorn W, Pakjaroen H, Thaikla K, Satheannoppakao W. Thai National Health
Examination Survey 2013-2014, NHES V [Internet]. 1st ed. Nonthaburi: Health Systems
Research Institute (HSRI); 2016 [cited 2021 Oct 31]. 283 p. Available from:
https://www.hiso.or.th/hiso/picture/reportHealth/report/report9.pdf

3. We change some of the references from manuscript version 1 as following lists:

 Ref no 31. was changed from “UNFPA Thailand. [Where do teen mothers live in
Thailand?]. [cited 22 May 2021]. Available from: https://thailand.unfpa.org/en/publications” to “Sukrat
B. Thailand Adolescent Birth Rate: Trend and Related Indicators. Thai Journal of
Obstetrics and Gynaecology. 2014 Jan 20;15–21.”. 

 The new reference was written in English, while the previous one was written in
Thai. However, the information was similar. 

 Ref no 34. changed from “Kim S, Ko Y, Lee HJ, Lim J. Menopausal hormone
therapy and the risk of breast cancer by histological type and race: a meta-analysis
of randomized controlled trials and cohort studies. Breast Cancer Res Treat.
2018;170: 667–675. doi:10.1007/s10549-018-4782-2. PMID: 29713854.” to
“Collaborative Group on Hormonal Factors in Breast Cancer. Type and timing of
menopausal hormone therapy and breast cancer risk: individual participant
meta-analysis of the worldwide epidemiological evidence. The Lancet. 2019
Sep;394(10204):1159–68. doi: 10.1016/S0140-6736(19)31709-X. PMID:
31474332.”. 

 The new reference is more relevant and more recent than the previous reference. 

In manuscript version 3, (revised from manuscript version 2; PONE-D-21-21989R1):

 We checked our references (from manuscript version 2). We revised the URL of
reference no.18 from “http://web.nso.go.th/en/census/poph/cen_poph.htm” to
“http://songkhla.old.nso.go.th/nso/project/search/index.jsp?province”
because the previous URL cannot access presently. However, the information is the
same as with the previous reference. 

 We revised the format of Ref no. 12, 13, 14, 40, 41, and 49 as the mark in
“Revised Manuscript with Track Changes” file.

Additional Editor Comments:

Suggestion: I would advise deleting "and encourage all oncologists and health
policymakers to manage breast cancer cases in Thailand." in line 20 as it seems not
the right place to make this call.

Line 119: April 20, 2020, October 18, 2020. —> April 20, 2020, to October
18, 2020.

Line 125: The status of the patients —> the vital status of the
patients

Line 128: The diagnosis date of cancer patients —> The date of
incidence.

Response: We already revised the text in line No. 20, 119, 125 and 128 following the
editor suggestion. (Revised version: Line 20, 119, 125, 129, and 130)

Suggestion: Lines 129-130: Because all these dates may not be the same, please, could
you be more specific on how you choose the date? The earlier available? 

Response: The incidence date was specified following the diagnosis date of cancer
patients in the cancer registry database. The date of incidence was obtained from
the most reliable source. The incidence date was firstly indicated as the date of
biopsy, if not available, the date when the specimen was received at the
pathological laboratory, or the report date. When the date of definitive
pathological diagnosis was not indicated, the date of hospital admission due to this
malignancy, radiological diagnosis, and other clinical diagnosis dates related to
the occurrence of malignancy or death were considered. We already revised the
paragraph. (Materials and Methods: Study setting and data sources, paragraph: 4)

Suggestion: Line 137: Does it means that the cancer registry can include patients
from other province too? If so, the population covered is not only that of the
Songkhla Province. Am I right?

Response: The Songkhla cancer registry is a population-based cancer registry. It
includes only the cancer patients who are citizens of Songkhla province. The
patients are identified from all hospitals in Songkhla province. Songklanagarind
hospital is a hospital in Songkla province. The cancer patients of Songklanarind
hospital who are the Songkhla citizen are registered to the Songkhla cancer registry
and the hospital-based cancer registry of Songklanagarind hospital. While cancer
cases, who are not Songkhla citizens, are not registered in the Songkla cancer
registry. They are registered only in the hospital-based cancer registry of
Songklanagarind hospital. We added more detail as “The hospital has been
running both hospital-based CR the Songklanagarind hospital and population-based CR
of Songkhla province. Subjects in this study were extracted from the
population-based cancer registry.” (Materials and Methods: Study setting and
data sources, paragraph: 5)

Suggestion: Line 431: what do you mean? Missing at random?

Response: Yes, we mean the missing at random. We already revised the sentence to more
clear as “…In the MICE process, the missing process does not violate
the missing at random assumption…”. (Discussion, paragraph 10)

Suggestion: Line 435: I am not sure to understand the link between this sentence and
the following one.

Response: Southern Thai women slightly differ from women in other regions regarding
genetic and cultural backgrounds. Though, we thought that the inference of this
study to all Thai women was still relevant with no solid evidence to validate
it.

Reviewers' comments:

Suggestion: 

Introduction:

Please add a reference to the first line “Breast cancer is the most common
cancer among women worldwide”.

Response: We already added the reference to that sentence. We wrote the sentence
based on reference number 1. (Introduction, paragraph: 1)

Suggestion:

Methods:

If you one method to determine malignancy is based on death, wouldn’t this
mean there may be an underestimation of the true prevalence of the disorder? I
assume not everyone who dies undergoes an autopsy. This should be addressed in the
limitations section.

Response: We would like to apologize not to address the diagnosis based on death
certificate in the limitation section.

 The percentage of the death certificate only (DCO) cases of Songkhla cancer registry
was reported around 0.5% in the Cancer Incidence in Five Continents (CI5). Usually,
the International Agency for Research on Cancer (IARC) doubts the quality of data
when the DCO is zero as the registry might collect cases from pathology sources
only. And none of the diagnosis was from autopsy. We are not sure how much pathology
sources only bias occurred in diagnosis of breast cancer in the registry. However,
the inclusion of cases diagnosed from death certificates ensures a better estimate
of the incidence. The cancer registries model the prevalence of cancer based on the
incidence and survival, not the count of existing cases in the registry. Since we
were estimating the incidence of breast cancer by receptor status, the presence of
DCO in cancer diagnosis would not affect the imputation process. 

Reference: the International Agency for Research on Cancer (IARC). Indices of data
quality of the Cancer Incidence in Five Continents (CI5) volume X. [cited 4 Feb
2022]. Available from: https://ci5.iarc.fr/CI5I-X/old/vol10/I_09.pdf

Suggestion:

You mention that your youngest case of breast cancer is in a women who is 24 years of
age, yet use 5 year age groups to calculate the age-specific rates. What is the
number of individuals who are 24 years of age? This group is likely very small, and
unstable. How does the estimate change if you collapse the 24 year old into the
upper age group?

Response: The number of individuals who are 24 years of age, were 2 cases from 2,883
cases. It estimated 0.1% of total cases. 

 The age-specific rates were calculated to the unit of cases per 100,000 population
in a particular 5-year age group according to Boyle P. & Parkin D.M. in which
age-specific rates were calculated from age group of 0–4 to 85+, which the
formula as following.

 a_i= (r_i⁄n_i )×100 000 

Where ai is an age-specific rate per 100,000 population in each five-year age group,
ri is the number of cases in the same five-year age group, and ni is the
corresponding person-years of the observation. 

 The ultimate aim of stratifying into five-year strata is to calculate the
age-standardized incidence rates (ASR) from the formula as follows:

 ASR= ( ∑_(i=1)^A▒〖a_i w_i
〗)⁄(∑_(i=1)^A▒w_i )

 Var (ASR) = (∑_(i=1)^A▒〖[a_i w_i^2 (100 000-a_i
〗)/n_i])⁄〖(∑_(i=1)^A▒w_i )〗^2 

 Where ai is an age-specific rate per 100,000 population in each five-year age group,
wi is world standard population in the corresponding five-year age group, and ni is
the person-years of sample population in the same five-year age group. 

 The variance of age-specific rate was still low in very young age groups in which
the number of patients were small, while the person-years in the sample population
were large, see Figure 1.

Reference: P. Boyle and D. M. Parkin, “Statistical Methods for
Registries,” In: O. M. Jensen, D. M. Parkin, R. MacLennan, C. S. Muir and R.
G. Skeet, Eds., Cancer Registration: Principles and Methods, IARC Scientific
Publication No. 95, International Agency for Research on Cancer, Lyon, 1991, pp.
126-158. 

Suggestion:

I was a little confused by this statement “including our projection of
age-specific incidence rates of the HER2- 194 positive subtype in 18 age groups
(0–4, 5–9, 10–14, ..., 80–84, and ≥85
years)”. How is this possible if you have the first case of cancer starting
at 24 years? Wouldn’t the relative cost for each of these other groups be
zero?

Response: The age-specific rates were calculated as described above. Even the
age-groups with no case have their associated person-years (ni) and the 95%CI could
be calculated by the following formula

ASR ± Z_(∝/2)× (s.e.(ASR))

s.e.(ASR)= √(Var (ASR)) 

The 95%CIs of age-specific rate in the very young age groups were narrow, since the
number of patients were small, while the person-years in the sample population were
large.

Suggestion:

Results:

Please include range of age in this sentence “The median age of the patients
at diagnosis of breast cancer was 53.0 years 237 (interquartile range, IQR:
46.0–62.0)”.

Response: We already included the age range in this sentence. The new sentence is
“The median age of the patients at diagnosis of breast cancer was 53.0 years
(interquartile range, IQR: 46.0–62.0; range: 24–95),…”
(Result: Participants and characteristics, paragraph: 2).

Suggestion:

Please also extend the table for age to show each 10 year age group from 20 to give
an idea of the instability in estimates for the younger age groups.

Response: We already revised table1 to show each 10 ages group, starting from 20-29,
30-39, until 70 and over. (Table 1)

Suggestion:

Figure 3 title says 105, please change to 100,000. Same in the sentence “Over
the study period, the ASRs of other subtypes were not greater than 5.5 cases per 105
women” And “. The ASR of this subtype in 2009 was 2.6 per 105
women.” And here “The overall ASR of HR-positive and HER2-positive
(HR+/HER2+ and HR-/HER2+) breast cancer increase from 30-32 and 10-11 to 39 and
13-14 per 105 women in 2020 to 2030”

Response: We did not find the sentence that that reviewer suggested to revise in the
second version (PONE-D-21-21989R1) of our manuscript. All of these sentences were in
the first version (PONE-D-21-21989)? We already revised all of these sentences.
(Figure 3 title, Result: Trends and projections of the age-standardized incidence
rates, paragraph: 2)

to Reviewers .docx
---

## [Editor Report · Decision Letter 2]

8 Feb 2022

PONE-D-21-21989R2

A rapid rise in hormone receptor-positive and HER2-positive breast cancer subtypes in
Southern Thai women: a population-based study in Songkhla.

PLOS ONE

Dear Dr. Sriplung,

There are some additional minor points I would like you to address before
accepting the manuscript for publication. Please see below.

Please submit your revised manuscript by Mar 25 2022 11:59PM. If you will need more
time than this to complete your revisions, please reply to this message or contact
the journal office at plosone@plos.org.
When you're ready to submit your revision, log on to https://www.editorialmanager.com/pone/ and select
the 'Submissions Needing Revision' folder to locate your manuscript
file.

If you would like to make changes to your financial disclosure, please include your
updated statement in your cover letter. Guidelines for resubmitting your figure
files are available below the reviewer comments at the end of this letter.

We look forward to receiving your revised manuscript.

Kind regards,

Sophie Pilleron, PhD

Academic Editor

PLOS ONE

Journal Requirements:

Additional Editor Comments (if provided):

Line 178: (0–4, 5–9, 10–14, ..., 80–84, and ≥85
years) —> 20-25, …, ≥85 as you don’t have cases under
20 years old.

Same comment for line 207 as you don’t have cases under 20 so it will be zero
anyway.

Lines 144-147: I appreciate that authors added details I suggested. However I would
suggest another formulation: Instead « The hospital has been running
both hospital-based of the Songklanagarind hospital and population-based cancer
registry of Songkhla province. Subjects in this study were extracted from the
population-based cancer registry. » I suggest: « In the
present study, we included only cases living in Songkhla province and registered in
the population-based cancer registry. »

Line 224: instead of « manage », I would suggest using
« impute ». This could give something such as
« we imputed missing values of the receptor status using MICE R
package. » I would also suggest mentioning clearly which variables you
used to impute.

Line 265-267: I would suggest deleting :  « Since the raw
dataset contained missing values for receptor status in approximately half of the
cases, the MICE package was used to assign values to the “unknown”
receptor status and classify the patients into four subtypes described in the Method
section. » as it is not a result per se and was already described in
the method section.

Line 441: I appreciate authors answered my comment. However, the sentence is not
correct. You cannot say for sure that missing data pattern is at random. You can
only assume it as it is not really verifiable. Author can consider something like:
« We imputed missing data assuming missingness was at random and we
properly modeled missing values. » instead of « In the
MICE process, the missing process does not violate the missing at random
assumption. »

Authors did not really answer the comment of Reviewer 2 regarding diagnosis made via
death certificate. The reviewer requested that authors acknowledge that some cancer
cases may have been missed since not everyone has an autopsy. In addition, I would
suggest authors to add %DCO in the method section where they mention it.
---

## [Author Response · Author response to Decision Letter 2]

1 Mar 2022

Response to Reviewers

comment on PONE-D-21-21989R3

We added the information of our data deposit in the Data Archiving and Networking
Services (DANS) in subtopic “Population denominators” and
“Management of missing data, paragraph 1”, and we added a reference
No. 19 in the manuscript as “19. Chuaychai, A. 1000 imputed data set of
receptor status for breast cancer. 2021. Data Archiving and Networking Services
(DANS). https://doi.org/10.17026/dans-xn5-5286”.

1. Please include captions for your Supporting Information files at the end of your
manuscript, and update any in-text citations to match accordingly. Please see our
Supporting Information guidelines for more information: http://journals.plos.org/plosone/s/supporting-information.

Response: We already added captions of the supporting information files at the end of
our manuscript. 

comment on PONE-D-21-21989R2

Additional Editor Comments (if provided):

suggestion:

Line 178: (0–4, 5–9, 10–14, ..., 80–84, and ≥85
years) —> 20-25, …, ≥85 as you don’t have cases under
20 years old.

Same comment for line 207 as you don’t have cases under 20 so it will be zero
anyway.

response: Yes, the cases under 20 year age group were zero cases in the result.
However, in the method part, we wrote 18 age groups, including age group under 20
years, because our inclusion criteria included all cases and age groups. That is why
we wrote 18 age-group in the methods part. We added the sentence “The count
of cases/ estimated Songkhla population in each 5-year age group might result in the
age-specific rate of zero in very young age groups.”. (Statistical analysis:
Incidence rates, trends, and projection, paragraph: 1)

suggestion:

Lines 144-147: I appreciate that authors added details I suggested. However I would
suggest another formulation: Instead « The hospital has been running both
hospital-based of the Songklanagarind hospital and population-based cancer registry
of Songkhla province. Subjects in this study were extracted from the
population-based cancer registry. » I suggest: « In the present study,
we included only cases living in Songkhla province and registered in the
population-based cancer registry. »

 response: We already revised following editor suggestion. (Study setting and data
sources, paragraph: 5)

suggestion:

Line 224: instead of « manage », I would suggest using « impute
». This could give something such as « we imputed missing values of
the receptor status using MICE R package. » I would also suggest mentioning
clearly which variables you used to impute.

 response: We already revised following editor suggestion. (Management of missing
data, paragraph: 1)

suggestion:

Line 265-267: I would suggest deleting : « Since the raw dataset contained
missing values for receptor status in approximately half of the cases, the MICE
package was used to assign values to the “unknown” receptor status and
classify the patients into four subtypes described in the Method section. »
as it is not a result per se and was already described in the method section.

response: We already revised following editor suggestion. (Proportions of breast
cancer M-IHC subtypes, paragraph: 1)

suggestion:

Line 441: I appreciate authors answered my comment. However, the sentence is not
correct. You cannot say for sure that missing data pattern is at random. You can
only assume it as it is not really verifiable. Author can consider something like:
« We imputed missing data assuming missingness was at random and we properly
modeled missing values. » instead of « In the MICE process, the
missing process does not violate the missing at random assumption. »

response: We already revised following editor suggestion. (Discussion, paragraph:
10)

suggestion:

Authors did not really answer the comment of Reviewer 2 regarding diagnosis made via
death certificate. The reviewer requested that authors acknowledge that some cancer
cases may have been missed since not everyone has an autopsy. In addition, I would
suggest authors to add %DCO in the method section where they mention it.

response: We added the information about the rules for reporting the incidence date
of population-based cancer registry as “The rules for reporting the incidence
date were according to the European Network of Cancer Registries (ENCR) [16], which
was also used by the Thai Cancer Registry Network and the International Agency for
Research on Cancer (IARC). The IARC allowed missing pathological diagnosis in
capturing cancer cases for population-based cancer registration as a low percentage
of patients migrating out of the captive area of a cancer registry usually occurred
in long-surviving diseases and death might occur before diagnosis in very short
surviving diseases.”. (Study setting and data sources, paragraph: 4)

to Reviewers_26022022.docx
---

## [Editor Report · Decision Letter 3]

2 Mar 2022

A rapid rise in hormone receptor-positive and HER2-positive breast cancer subtypes in
Southern Thai women: a population-based study in Songkhla.

PONE-D-21-21989R3

Dear Dr. Sriplung,

We’re pleased to inform you that your manuscript has been judged
scientifically suitable for publication and will be formally accepted for
publication once it meets all outstanding technical requirements.

Kind regards,

Sophie Pilleron, PhD

Academic Editor

PLOS ONE

Additional Editor Comments (optional):

The manuscript would deserve to be check for English language.
---

## [Editor Report · Acceptance letter]

9 Mar 2022

PONE-D-21-21989R3 

A rapid rise in hormone receptor-positive and HER2-positive breast cancer subtypes in
Southern Thai women: a population-based study in Songkhla. 

Dear Dr. Sriplung:

I'm pleased to inform you that your manuscript has been deemed suitable for
publication in PLOS ONE. Congratulations! Your manuscript is now with our production
department. 

Kind regards, 

on behalf of

Dr. Sophie Pilleron 

Academic Editor

PLOS ONE